# Biomolecule-Based Optical Metamaterials: Design and Applications

**DOI:** 10.3390/bios12110962

**Published:** 2022-11-02

**Authors:** Ana Laura Torres-Huerta, Aurora Antonio-Pérez, Yolanda García-Huante, Nayelhi Julieta Alcázar-Ramírez, Juan Carlos Rueda-Silva

**Affiliations:** 1Escuela de Ingeniería y Ciencias, Tecnológico de Monterrey, Campus Estado de México, Av. Lago de Guadalupe KM 3.5, Margarita Maza de Juárez, Cd. López Mateos, Atizapán de Zaragoza 52926, Mexico; 2Departamento de Ciencias Básicas, Unidad Profesional Interdisciplinaria en Ingeniería y Tecnologías Avanzadas, Instituto Politécnico Nacional (UPIITA-IPN), Mexico City 07340, Mexico; 3Department of Genetics, University of Cambridge, Cambridge CB2 3EH, UK

**Keywords:** biomolecule-based metamaterials, nanostructure, nanoparticles, hydrogel, crystals, lattices

## Abstract

Metamaterials are broadly defined as artificial, electromagnetically homogeneous structures that exhibit unusual physical properties that are not present in nature. They possess extraordinary capabilities to bend electromagnetic waves. Their size, shape and composition can be engineered to modify their characteristics, such as iridescence, color shift, absorbance at different wavelengths, etc., and harness them as biosensors. Metamaterial construction from biological sources such as carbohydrates, proteins and nucleic acids represents a low-cost alternative, rendering high quantities and yields. In addition, the malleability of these biomaterials makes it possible to fabricate an endless number of structured materials such as composited nanoparticles, biofilms, nanofibers, quantum dots, and many others, with very specific, invaluable and tremendously useful optical characteristics. The intrinsic characteristics observed in biomaterials make them suitable for biomedical applications. This review addresses the optical characteristics of metamaterials obtained from the major macromolecules found in nature: carbohydrates, proteins and DNA, highlighting their biosensor field use, and pointing out their physical properties and production paths.

## 1. Introduction

Metamaterials exhibit physical properties such as negative electrical permittivity and magnetic susceptibility not obtainable from natural materials, thanks to their engineered microarchitecture [1,2]. These characteristics are derived from the material’s structural pattern rather than from its intrinsic properties [2], making metamaterials one of the most prominent types of engineered materials and allowing the production of highly tunable devices and sensors [1].

Of special interest are optical metamaterials that can alter electromagnetic waves at determined optical frequencies, conferring properties such as high transparency, high light absorbance, negative refractive index, hyperbolic dispersion and absorption of light at both a broad range of wavelengths in the electromagnetic spectrum or at precise wavelengths [1,3,4,5,6,7]. Proposed optical metamaterials applications include devices with integrated X-Ray or UV light sources, highly-sensitive sensors, cloaking, improved photovoltaic cells, and light-tunable control mechanisms for electromagnetic systems [1,4,5,6,7]. These devices could be applied in multiple areas such as lab-on-a-chip systems, microfluidics, point-of-care diagnostics, biological system sensing and regulation, health monitoring, light-controlled therapeutics, high-resolution and label-free imaging, drug delivery, and tissue engineering [3,8,9,10]. On the other hand, biomolecules exhibit chirality, a property that refers to the structure attribute of an object without a mirror plane or inversion symmetry. It is very important for the chiral configuration for biomolecules in playing their functions. Depending on the chirality of the biomolecules, their interaction with the spin of photons results in circularly polarized light that will be left- (LH) or right-handed (RH) [11]. Although proteins and DNA only exhibit weak optical activity in the UV range, their combination with plasmonic nanoparticles changes the circular dichroism response in the UV region [12,13,14]. Interestingly, chirality plays a very important role in biosensors made with plasmonic materials that are functionalized with biomolecules [15]. New biomolecule-based chiral metamaterials are promising to have superior resolution and a wide variety of plasmonic assemblies. Additionally, this kind of materials can realize negative refractive index, light polarization control, reconfigurable chirality, stimuli-responsive behavior, and chirality sensing [11,16,17,18].

The use of biomolecules for optical metamaterial engineering guarantees the compatibility of these devices with biological environments. In addition to their biocompatibility, biomolecule-based metamaterials also offer other advantages, such as high tunability through external stimuli, low-cost, simple, and scalable production without compromising performance [3]. For example, Salim and Lim [19] obtained engineered biosensors with high sensitivity, enhanced limit-of-detection and high resolution. Another potential application of biomolecule-based metamaterials lies in the development of functional 3D scaffolding for tissue engineering [20]. Additionally, the use of biological metamaterials has been shown to have potential for the creation of implantable devices, useful for patient health monitoring and controlled drug-delivery [8].

The present review aims to provide a comprehensive overview of the application of different biomolecules in the design of novel metamaterials and review different techniques that have been proposed for the mass scale production of these biomaterials, depending on their intended application. This review provides an insight of the different designs that have been proposed for the creation of novel optical metamaterials based on polysaccharides, peptides and nucleic acids. The document also addresses the evaluation of the results obtained in previous works using these materials, their advantages and the challenges that need to be addressed before the widespread implementation of them. This review aims to provide the biological insights required for the use of biomolecules in both the production and implementation of metamaterials.

The first section of this review discusses the advantages of the use of polysaccharide matrices in the construction of optical metadices. The second section focuses on the use of proteins for the creation of solvent-free protein liquids, metafluids, metasurfaces, protein templated metamaterials and hydrogels aboarding its potential for the design of optical metamaterials. The third section examines the use of nucleic acids as raw materials for the assembly of metamaterials and the proposals for the functionalization of nucleic acid surfaces with metal nanoparticles.Finally, the last section explores methodologies for biomolecules production, highlighting their advantages over traditional metamaterials in terms of scalability, cost and ease of production.

## 2. Photonic Biomaterials Found in Nature

In biopolymers, some properties such as light propagation, scattering and emission are controlled through hierarchical and chemical structures (Figure 1). These can be manipulated to synthesize a wide range of optically active nanocomponents with optical traits as iridescence, or chiral photoluminescence and with advantages such as mechanical robustness, renewability, biocompatibility and ambient processing conditions [21].

### 2.1. Main Sources to Obtain Polysaccharides

#### 2.1.1. Cellulose

Cellulose, the most abundant polysaccharide in the world, is present in lignocellulosic material together with hemicellulose and lignin. In the last few years, exploitation of vegetal biomass to obtain value-added products such as biofuels, chemicals and advanced materials has taken a lot of interest. However, lignin limits the utilization of materials such as lignocellulose feedstocks. Hence, many pretreatment methods focus on o removing lignin from lignocellulose feedstocks or increasing cellulose exposition. The efficiency of these methods depends on the content of cellulose, hemicellulose, lignin, water, etc. and the pretreatment must be carefully selected in function of the vegetal biomass properties (Figure 2). Recently, Ning et al. [22] described several current lignocellulose-pretreatment methods, addressing their advantages and disadvantages. We highly recommend this review for diving deeper into the subject.

Besides lignocellulose feedstock, bacterial culture is another source of cellulose. Known as bacterial cellulose (BC), it possesses a unique nanofiber-weaved three-dimensional reticulated network structure, that provides excellent mechanical properties, high water holding capability and exceptional suspension stability. Compared to vegetal cellulose, BC has a higher purity, as it does not contain lignin nor other components. In addition, BC has a high crystallinity degree, as well as biocompatibility and biodegradability [23]. *Gluconacetobacter*, *Aerobacter*, *Rhizobium*, *Sarcina*, *Azotobacter*, *Agrobacterium*, *Pseudomonas*, and *Alcaligenes* are among the bacterial genera that can produce BC [24], being *Gluconacetobacter xylinus* (previously named Acetobacter xylinus) the most widely studied microorganism to produce BC [25]. Its biosynthesis occurs in two stages: glucose intracellular polymerization into cellulose polymers and self-assembly of cellulose polymer chains into crystalline nanofibers [26].

#### 2.1.2. Alginate

Alginate is a linear polysaccharide with different applications in the food and pharmaceutical industries. It has a chemical structure composed of subunits of (1-4)-β-d-mannuronic acid (M) and its C-5 epimer of α-l-guluronic acid (G). Monomer composition and molecular weight dictate alginate properties [27]. Alginate production comes mainly from brown seaweed extraction under alkaline conditions. The process includes washing the algae at acidic conditions, then performing extraction with a Na_2_CO_3_ solution at 80 °C and pH 10. Later, the solution is filtered, and the alginate is precipitated by adding CaCl_2_. The obtained alginate fibers are then treated with HCl to produce alginic acid and neutralized using Na_2_CO_3_, thus obtaining sodium alginate. Finally, the product is dried with hot air, milled and screened at different sizes [28].

Although alginate is produced commercially from algae, it also can be synthesized as an extracellular polymer through a bacterial culture of *Azotobacter vinelandii* and *Pseudomonas* spp. This method provides tight control on properties such as molecular weight and physicochemical characteristics by adjusting culture conditions during fermentation [27].

#### 2.1.3. Chitosan

Chitosan is a linear polysaccharide, bio-derivative of chitin, naturally existent in shells of crustaceans, jellyfish, corals, insects, and cellular walls of fungi [29,30]. This polysaccharide has anti-inflammatory, antimicrobial, antitumor, and immunity-enhancement properties, with a broad variety of purposes in biomedicine [29]. Various techniques to extract chitin from crustaceans have been reported in the literature; the employed techniques depend on the characteristics of the source material, since its composition differs considerably from one species to another. The chitosan obtention process consists mainly of four steps: demineralization, deproteinization, discoloration, and deacetylation. Most of the reported techniques rely on chemical processes of protein hydrolysis and the removal of inorganic matter [31].

The main goal of the demineralization step is to eliminate the CaCO_3_ contained in the crustacean shells by diluting them in solutions of up to 10% of HCl at room temperature, depending on the chitin source, other acids such as HNO_3_, HCOOH, H_2_SO_4_, and CH_3_COOH can be used to perform this step.

To deproteinize the shells, they are soaked in a NaOH solution and heated at 65 to 100 °C, from 0.5 to 72 h. However, long-time and high-temperature treatments might cause chain rupture and partial deacetylation of the polymer. Besides NaOH, other chemical agents such as Na_2_CO_3_, NaHCO_3_, KOH, K_2_CO_3_, Ca(OH)_2_, Na_2_SO_3_, NaHSO_3_, Na_3_PO_4_ and Na_2_S have also been utilized.

The discoloration step involves the solvent extraction of the pigments found in the shells, with acetone, chloroform, ether, ethanol, ethyl acetate or solvent mixture at room temperature or with traditional oxidizing agents.

Finally, chitin is deacetylated by hydrolysis of the acetamide groups in concentrated solutions of NaOH or KOH (30–50%), at temperatures above 100 °C, either in an inert atmosphere or in the presence of reducing substances [31].

Table 1 shows examples of carbohydrate-based structured materials, their optical properties, and their obtention method.

### 2.2. Polysaccharide-Based Optical Metamaterials

Polysaccharides are biopolymers formed by repeating units joined by glycosidic linkages, their general formula is ((C_6_H_10_O_5_)*_n_*, where *n* can obtain values in the range of 40 to 3000 [39,40]. Their major obtention source is from vegetal biomass; however, biopolymers as chitin -a chitosan derivative-, can be found in the cell wall of fungi, shells of crustacean, jellyfish, corals and insects [29]. Polysaccharides will present different chemical compositions, molecular weights, chemical structures, surface properties and extensive networks of intra- and inter-hydrogen bonding depending on the source from which they were obtained [39,40,41].

Given their natural origin, polysaccharide-based metamaterials possess appealing characteristics such as biocompatibility and biodegradability, making them suitable for many practical uses such as packaging, electronics, and pharmacology [42]. In addition to this, polysaccharides exhibit remodeling capabilities, high transparency, and light guiding efficiency. Therefore, they have an undeniable potential use in the fabrication of optical materials with a wide range of purposes including energy harvesting, responsive structures to any stimuli, biosensor implantations, scaffolds for repairing damage tissues, etc. Polysaccharides have been designed in a fashion that allows them to interact with the analyte through different ways, such as chelation, electrostatic interaction, and hydrogen bonding [39]. The main polysaccharides utilized for manufacturing biosensors include cellulose, chitosan, agarose, starch, and others [43].

Nature represents an extraordinary source of inspiration for engineering materials, and the design of materials with optical properties is not the exception. For example, how the structural conformation adopted by the pigmented epidermal cells dictates the reflection intensity of petal flowers [44]. Another instance of nature-inspired optical-structured material is found in the growth of multilayered and structured photonic crystals, responsible not only of the blue butterfly wings’ iridescence, but also it repels dust and water and assists in the thermal regulation of the insect. Such convergence of function has devised the optofluidic discipline, which combines the photons flow in macro- or microfluidic channels to modulate and manipulate the light [45]. The study of the disorder in several nanostructured materials, has also led to the creation of optical materials with interesting and useful properties. Thus, the comprehensive knowledge of biological photonic structures enable us to control light propagation, scattering, and emission via hierarchical structures and diverse chemistry [21,46].

#### 2.2.1. Cellulose-Based Optical Material

Cellulose is the most abundant polysaccharide on the Earth, composed of hundreds to thousands of β 1,4-linked D-glucose units [21]. Cellulose is a suitable biodegradable material for biosensors, given its transmittance of visible light and good permeability for water and ions. Cellulose can react with different chemicals, making it favorable for fabricating microstructure and biodegradable materials with optical, microfluidic, and drug release performances [43].

In nature, the iridescence shown by leaves and some fruits is produced either by multilayer interference of two or more materials with different refractive index, or by multiple microfibril layers organized in a special way, forming a helicoid structure that seems like a liquid crystal nematic phase (Figure 1a) [44].

Cellulose derivatives can replicate photonic structures. For example, Caligiuri et al. [47] fabricated microscale photonic structures and nanoscale metasurfaces employing pure cellulose and agro-wastes as a replica-molding polymer. The cellulose films showed a refractive index and transparency that was well-suited for photonic structures operating in the visible-near-infrared (vis-NIR) spectrum. In addition, the authors overcoated the nanostructured cellulose films with Ag, so they formed a 2D plasmonic disk-hole matrix (Figure 1b).

Several bacterial strains can also produce cellulose as a structural part of the biofilms, Caro-Astorga et al. [48] reported a protocol to produce bacterial spheroids from an engineered *Komagataeibacter rhaeticus* culture. These were used as building blocks to produce 2D- and 3D- shapes of engineered living materials, containing genetically functionalized bacteria that impart fluorescence, while it sends and receives signals. By combining more than 20 different growth variables, they observed that the initial optical density and the size of the culture container were two critical factors that affect the bacterial cellulose spheroid formation. Thus, this research enhances the understanding of bacterial cellulose spheroids formation, and paves the way for the design of biomaterials that can be used in tissue regeneration.

On the other hand, Cai et al. [49] fabricated cellulose-based microspheres as basic building blocks using cellulose nanofibrils (CNF) by spray-freeze-drying method. The microspheres had pore sizes ranging from nano- to micrometer order, the cross-links formed by the nanofibrils allowed the obtaining of stable aerogel microspheres even in harsh conditions. This method did not use toxic solvents nor required a purification process. As a consequence, the obtained nanofibril aerogel microspheres were suitable as cell culture scaffolds because they were nontoxic, biocompatible, and showed interconnected nanofibrous with high porous structure.

Bacterial cellulose (BC) membranes also display peculiar and unique properties that allow their utilization from biomedicine to photonics applications. Santos et al. [35] fabricated novel thermal/electrical responsive photonic composite films by combining the low molecular weight nematic liquid crystal, 4′-(hexyloxy)-4-biphenylcarbonitrile (HOBC) and bacterial cellulose nanocrystals (CNC), resulting in films that integrate iridescence, conductive and thermal properties. In addition, the films showed chiral nematic organization confirmed by the scanning electron microscopy (SEM). This report represents a method for fabricating composites with attractive properties to the development in the field of thermo- and electro- optical devices, such as smart windows, optical sensors, and display devices.

Furthermore, a green slab waveguide for plasmonic sensors based on BC was fabricated by Cennamo et al. [50] via sputtering gold on the top of a BC-paper composite.The localized surface plasmon resonance (LSPR) showed a response to refractive index changes when tested with different water-glycerin solutions, thus representing a good alternative to eco-friendly and disposable biosensors.

In 2016, Rull-Barrul et al. [51] fabricated a portable, recyclable and highly selective paper-based sensor device for the colorimetric and optical detection of hydrogen sulfate in water. The device changes from deep pink to pale rosy depending on the analyte concentration and allows a semiquantitative estimation of the sample concentration.

From data communication to sensors, optical fibers are present in our daily life. These coaxial structures are generally made from glass or plastic. Despite their excellence for communication applications, they represent a challenge to modify its material for sensor applications [52]. In addition, optical fibers show several advantages as chemical and biological sensors such as being cost effective, resistant to electromagnetic interference and applicable in situ. Furthermore, its structure can be manipulated in diverse ways to produce different sensor performances [39]. In this regard, it is possible to perform a surface plasmon resonance technique using optical fibers in two different ways. One of them uses a d-shaped optical fiber in which the cladding is removed entirely from one side of the fiber (Figure 1c), whereas in the second way the cladding is not removed entirely, thus just a small region of the fiber surface is polishing. By applying a metallic thin film from Au or Ag, it is possible to initiate a plasmonic effect [39]. In 2020, Orelma et al. [52] studied the preparation of an optical cellulose fiber from two-different types of cellulose, which were regenerated cellulose and cellulose acetate, whose refractive index were higher and lower, respectively. Hence, an optical cellulose fiber to sense water was prepared by dissolving cellulose in [EMIM]OAc, which was dry-wet spun into water. By coating a layer of cellulose acetate, the cladding layer on the cellulose core was obtained through its dissolution in acetone, using a filament coater. The optical fiber was able to guide light in the range of 500–1400 nm, showing an attenuation constant of 6.3 dB/cm at 1300 nm, given by the cellulose fiber. Thus, a contact water-based response was observed as a clear attenuation in the light intensity. Therefore, authors suggest this optical fiber might be used in sensor applications [52].

Liquid crystals are a state of matter intermediate between that of a crystalline solid and an isotropic liquid. They possess many of the mechanical properties of a liquid, such as high fluidity, formation, and coalescence of droplets. At the same time, they exhibit anisotropy in their optical, electrical, crystal-like, and magnetic properties [53]. Cholesteric liquid crystal (CLC) is the phase that possesses intrinsic periodicity in the form of helical supramolecular structure. They combine optical anisotropy properties that are inherent for crystals and mobility. All of this makes CLC highly useful for the creation of diffraction grating (DG) with tunable properties [54]. Dai et al. [55] designed a simple, green, cost-effective colorimetric ammonia gas sensor based on cholesteric liquid crystal films of copper(II)-doped cellulose nanocrystals (CNC-Cu(II)). The denominated CNC-Cu(II)125 film was sensitive to ammonia gas and produced a red-shift of reflective wavelength as well as an effective colorimetric transition.

#### 2.2.2. Chitin-Based Optical Material

Chitin is a natural bio derivative polysaccharide found mainly in shells of crustaceans, jellyfish, and corals. Chitosan, is a linear, hydrophilic polymer, with important properties such as nontoxicity, natural biodegradability, and biocompatibility, which make it suitable for biomedical approaches [29]. Its vastly hydroxyl and amino side groups, as well as its enhanced film forming properties, grant reversible water adsorption and tremendous binding to noble and transition metal ions, what alters its optical properties and makes it a candidate for sensors with high potential [29,43]. Additionally, chitosan swells by absorbing water molecules from the air, and several biosensors have been constructed based on this principle [43,56]. Using its absorption properties, Jang et al. [57] created a self-powered humidity sensor made by a metal-insulator-metal (MIM) configuration based on a Fabry-Pérot resonator. The insulator part was a chitosan hydrogel sandwiched between two silver layers and combined with a photovoltaic cell, resulted in a dynamic and real-time colorimetric humidity sensor. Therefore, the MIM configuration system represents a potential for the development of self-powered sensors. In addition to this, a spin Hall effect-based platform using chitosan-coated all-dielectric metamaterial was proposed to detect relative humidity. This system demonstrated to be a promising approach for detecting biomolecules that do not interact with electromagnetic waves in the UV region. Besides their higher biocompatibility, biodegradability and hydrophilicity, these sensors are advantageous by being mass-produced using nanoscale manufacturing methods [11].

Photonic crystal hydrogels (PCHs) consist of both periodic photonic crystals (PCs) and stimuli-responsive hydrogels and can act as a sensing system for the detection of specific analytes. Therefore, they have gained high interest in sensing applications. In PCHs the hydrogel system experiments volume phase transitions (swelling or shrinkage) upon exposure to an external stimuli, which results in a change in the photonic stopband of the PCs, and in a shift of the Bragg diffraction peak wavelengths [58].

As a branch of PCs, opal PCs have been widely adopted in the label-free detection based on their broad band gap and structural color, whereas inverse opal photonic crystals (IOPC) refer to the polymer films with the regular voids, and they are a class of functional materials that exhibit various vivid diffraction signals by varying external stimuli [59,60,61]. The volume of periodic building blocks of IOHGs can vary through the change of effective refractive index or the lattice parameter, which endow the emergence of bright colors and allow visual detection [7,62,63,64], Huang et al. [65] built inverse opal films from chitosan, responsive to different organic solvents such as ethanol, methanol and 1-propanol, important solvents in pesticide and pharmaceutical production, showing the potential application of biodegradable chitosan-based inverse opal films with reversible color change [65].

#### 2.2.3. Agarose- and Other Polysaccharides-Based Hydrogels

Agarose uses a span of applications, from DNA separation to tissue regeneration. This polysaccharide derived from agar shows a high potential for incorporating biological subjects to biosensing applications. The optimization of its optical properties can be reached by testing different concentrations and tuning their optical refractive index as performed in agarose hydrogels. For example, hydrogel/cell hybrid optical waveguides with a light propagation loss of 12–13 dB/cm, approximately, have been developed for the encapsulation of living cells in agarose gel [43,66].

Although optical sensors are appealing detection platforms for the quantification of a specific analyte, they are not fully compatible with biological systems for implantation in vivo, making necessary the development of biocompatible implantable biosensors [67]. Nevertheless, carbohydrate-based hydrogels have found biomedical application due to their biocompatible and biodegradable nature, as well as their tunable optical and mechanical properties [67,68].

In this regard, Yetisen et al. [67] fabricated hydrogel optical fibers for continuous glucose sensing in real time, which consist of poly(acrylamide-co-poly(ethylene glycol) diacrylate) cores functionalized with phenylboronic acid. The complexation of the phenylboronic acid and cis-diol groups of glucose enables reversible changes of the hydrogel fiber diameter. The analysis of light propagation loss allowed quantitative glucose measurements within the physiological range. Hydrogels can be functionalized with several elements such as chelating agents, proteins, oligomers, etc., making them responsive to a wide range of analytes for sensing and drug delivery applications. Cai et al. [69] developed a responsive photonic crystal hydrogel sensor containing lactose, galactose and mannose to detect lectin proteins, based on the specificity and affinity of protein-carbohydrate interaction, which can be highly improved by the clustering of multiple carbohydrates. Moreover, the optical properties of hydrogels by patterning host materials can be programmed. In 2019, Hess et al. [70], designed a gel showing polarization-dependent plasmonic properties, with coaligned gold nanorods and cellulose nanocrystals through 3D-printing methods. Additionally, hydrogels also have important applications for mimicking human tissue characteristics. Recently, a hydrogel that mimics the cartilage extracellular matrix was developed based on a photo cross linkable alginate bioconjugated with both gelatin and chondroitin sulfate, nanocomposited with liquid graphene oxide. The hydrogel boosted the proliferation of human adipose tissue-derived mesenchymal stem cells (hADMSCs) making it a promising scaffold material for tissue engineering [71].

Another type of polysaccharide-based materials are aerogels, which are unique solid-state materials composed of interconnected 3D solid networks and many air-filled pores that combine the structural and physicochemical properties of nanoscale building blocks to macroscale integrating characteristics such as large surface area and low density. These features endow aerogels with high sensitivity, high selectivity, as well as a fast response, and recovery for sensing applications [72]. Not long ago, Druel et al. [73] fabricated starch-based aerogels, observing that starches containing low amylose amounts produced aerogels with higher density and lower specific surface area.

The bio-based nature and remarkable features exhibited by polysaccharides has allowed their use in optical sensor applications [39]. For instance, it has been showed that spiral nanostructures, have the ability to excite surface plasmon fields endowed with orbital angular momentum [74]. Additionally, it has been described that functionalized cellulose and amylose have yield novel chiral functions as asymmetric organocatalysts, chiral auxiliaries and chiral fluorescence sensors. Additionally, it has been demonstrated the incorporation of unique features to these biopolymers have enabled the design of circularly polarized luminescence materials due to the chirality showed by the moieties by which they are conformed [75]. However, improvements on the long-term stability of these materials have to be addressed for the commercialization of polysaccharide-based sensors.

## 3. Protein Based Metamaterials

Normally, the elements used in the construction of optical metamaterials possess finely tuned chemical and physical properties to perform a specific, unique and static function. Conventional metamaterials are often limited due to their constituent materials’ rigidness, which usually lack flexibility and tuning capability. However, the adaptation of optical devices to fuse within electronic devices or biomedical platforms has extended its scope to a variety of adaptive responses and the need for intrinsic dynamic behavior [76]. Under this evolutionary context, the application of peptide macromolecules or proteins in the construction of optical metamaterials has been approached with greater frequency.

Proteins have multiple levels of organization and the greatest structural diversity of all the macromolecules in nature. Amino acids that form the polypeptide chain lead to tridimensional structures with precise control of nanomorphology [77]. Additionally, the functional groups present in a protein sequence lead to controlled conformational transitions and dynamics as a specific response to external stimuli, making proteins biomolecular elements with potential in the development of metamaterials. A plethora of native proteins with distinguishing mechanical, chemical, electrical, electromagnetic and optical properties has been studied [78]. Elastins, collagens, actines, silks, keratins and reflectins are some of the proteins considered for protein-based biomaterials and soft materials in metamaterials with tuning capability [79].

### 3.1. Solvent Free Protein Liquids

Liquid crystals are defined as the fourth state of matter forming between solid and liquid states exhibiting both the fluidity of a liquid and the spatial order found in solid crystals. The variety of phases depends on temperature or solvent changes (thermotropic or lyotropics, respectively) [80]. Liquid crystals contain subwavelength nanostructures with unconventional bulk optical properties and are one of the more common soft materials applied in metamaterials due to its numerous helpful optical and physical properties [3,81]. Proteins, as biomacromolecular components, have many functional groups governed by forces such as van der Waals, ionic interactions, or hydrogen bonds that allow different physical states [82]. Additionally, proteins exhibit the great advantage of long-term natural evolution, rendering a wide variety of biological, chemical, and physical functions [83]. Despite all the diversity of interactions that stabilize protein structures as nanoscale objects, proteins exhibit persistent structures with dimensions that exceed the range of their intermolecular forces, making liquid–vapor coexistence unattainable. On the other hand, the preparation of highly concentrated protein solutions with conserved functionality is hard, due to the aggregation tendency of proteins [84]. Nonetheless, in 2009, Perriman and coworkers successfully combined a ferritin with surface modifications that extended the range of intermolecular interactions to an anionic polymeric surfactant. The nanoconstruct was formed as a single-component, stoichiometric conjugate by electrostatic grafting of an anionic polymer surfactant with a central polyethylene glycol moiety to surface-accessible cationized amino-acid residues. This work was the first reporting a solvent-free protein melt method with zero vapor pressure to produce a stable viscoelastic protein liquid that exhibits thermotropic liquid-crystalline behavior with intact protein structure and functionally at 30 °C [85]. The same research group also developed a similar electrostatically-complexed myoglobin-based solvent-free liquid protein, with native globular structure retention and active metallocenter functionality, capable of reversible non-cooperative dioxygen binding in the solvent-free state [86].

Another example of the application of proteins for the creation of liquid crystal-type materials was the one developed by Liu et al. [87] using GFP (Green Fluorescent Protein). GFP, from *Aequorea victoria* jellyfish, is able to emit green light in the process of bioluminescence. However, further findings together with GFP forced mutations, gave the family of GFPs the ability to cover the whole visible spectrum with their emissions. In this work, Liu et al. [87] genetically engineered a series of solvent-free GFP and functionalized elastin-like protein liquid crystals and liquids by electrostatic complexation of supercharged polypeptides with surfactants containing flexible alkyl chains. GFP was fused as a “model protein” to these supercharged polypeptides. By modifying the extension of the aliphatic chains on the surfactant structure, they changed the intermolecular forces in the complexes with broad temperature range control. Remarkably, the fluorescence properties of GFP were maintained during the preparation process, indicating that the folded protein maintained its integrity during the surfactant treatment. This work also demonstrates that other folded proteins can be “liquefied” using this method. This “new” generation of solvent-free fluidic materials, termed as “functional molecular liquids” (FMLs). These hybrid configurations containing solvent-free liquid biological molecules will open new exploration paths for novel functions of nanomaterials in the biotechnology field [84].

### 3.2. Protein Based Metasurfaces

Metasurfaces are structures that consist of optical components patterned on an ultrathin flat surface with subwavelength dimensions. The building blocks of metasurfaces are the meta-atoms on a subwavelength scale. Metasurfaces can shape the wavefront of light to introduce a desired spatial profile of the optical phase by modifying the pattern or optical properties of its atomic optical components. To confer various optical functions to the metasurfaces and diversify their optical properties, dielectric materials and different meta-atoms have been used [88]. Proteins provide an interesting alternative for the synthesis of smart surfaces and could lead to new applications, including bioelectronics, bio optics and biophotonics due to their dynamic behaviors, reconfigurable higher level assembly, and tunable physical properties when receiving external stimuli [77,89,90]. These versatile building blocks have an inherent structural and functional complexity that is unmatched by synthetic materials [91]. The integration of proteins with micro/nanopatterning strategies make possible a number of fascinating smart surfaces, where on-demand tuning of structure or function can enable new application opportunities for biologically compatible, protein-based micro/nanopatterned materials. During the last decade, a large number of strategies for protein pattern have been developed, such as micro-contact printing, inkjet printing and optical, interferometric or electron beam lithographs, among several other techniques [92].

Proteins are broadly classified as fibrous and globular proteins. Fibrous proteins usually play important structural roles, providing external protection, support, shape, and form [93]. Structurally, fibrous proteins consist of elongated polypeptide chains that run parallel to one another, are stabilized by cross-linkages, and contain periodic distributions of their charged and/or apolar residues [94]. Examples of fibrous proteins are collagen, α-keratin, keratin, elastin and silk. Silk is a fibrous protein made by silkworms, spiders, and other insects. Each insect produces silk components with a different amino acid composition translating into different structural properties. Silk from silkworm is composed of two different proteins, sericins as “glue protein” and fibroin as “core protein” [95]. Silk fibroin is an amphiphilic block copolymer with a heavy chain composed of 12 repetitive domains predominated by the sequence G-X-G-X-G-X (G = glycine 43%; X = alanine 30% or serine 12%). Other amino acids such as tyrosine (5%), valine (2%), and tryptophan are present in smaller proportions [96]. The primary structure of silk fibroin of silk worm contains 12 repetitive hydrophobic domains separated by 11 hydrophobic non-repetitive regions. Silk biopolymers can be reshaped on the micro and nanoscale through the addition of solvent or chaotropic agents solutions, inducing aggregation [97]. This property allows silk fibroin to be used for 3D printing or other protein pattern techniques. Reshaped and patterned silk proteins at visible wavelength scale in combination with the favorable bulk optical properties of silk films, can offer a platform for the production of micro and nano patterned silk optical elements.

In 2009, Amsden et al. [98] developed a periodic nanopatterned 2D lattice in pure silk fibroin films with the ability to maintain biological activity within the silk optical elements, by simply mixing biological dopants in silk fibroin solution to form optical elements. Silk nanostructures were created by nanoscale electron-beam writing on hard metal masks (Cr metal on Si wafers, illustrated in Figure 3a). Nanopatterned silk structures were illuminated with visible light and scattered light in different colors according to lattice spacing, angle of illumination and angle of collection.

The research developed by Liu et al. [99], confirmed the versatility of silk protein for manipulation and structuring. They synthesized gold nanoparticles in-situ by heating the silk fabric immersed in gold ion solution, with HAuCl_4_ at different concentrations (0.1–0.6 mM). Metal nanoparticles assembled or synthesized on silk fibers lead to different captivating colors on the surface of the treated silk, also giving rise to enhanced electromagnetic fields for amplification of optical signals. The observed properties acquired depended on the gold content in the silk fabrics and the morphologies of the gold nanoparticles on the silk fabric. According to these results, the different refractive indexes of silk and the aqueous solution lead to change in the optical properties of the gold NPs, suggesting a capability of these films to be used as biosensors.

Various strategies have previously been developed to obtain sliced or thin film materials with high transparency from silk fibroin [100,101]. However, some of these methods have very low efficiencies and use environmentally-unfriendly chemical compounds [102,103]. By a novel and sustainable cyclic high-pressure fibrillation method, Okahisa et al. [104] produced a “silk fibroin paper” with high transparency, high crystallinity, high elastic modulus, and high thermal stability, using only water. They treated silk from *Bombyx mori* to eliminate the sericin protein and obtain fibroin. The degummed fibroin was ground and subsequently homogenized by grinder MKCA6-3, Masko Sangyo Co. and high-pressure water-jet nanofibrillation (WJ) system, respectively. The configuration of this homogenization system allows the fibrillation of diverse materials up to dimensions of 15–25 nm. The suspensions of fibroin were homogenized 1, 9, 15, and 30 times. Then, each silk fibroin “paper” was obtained by filtration and drying. The obtained silk fibroin paper thickness varied from 75 μm to 55 μm according to the number of homogenization passes (1 pass to 30 pasess). Additionally, the optical transmittance evaluated at 600 nm increased with the number of passes on the homogenizer, from 47.9% to 0 passes until 82.7% to 30 passes, which means that the transparency increased as a more exhaustive mechanical fibrillation treatment was performed. It seems that there is a limit on the nanofibrillation of fibroin by mechanical treatment only. It is important to note that these features were achieved without the addition of organic solvents or toxic chemicals.

Additionally, after years of evolution, there are proteins in nature with photonic nanoscale architectures that can interact with light via interference, diffraction or scattering. Reflectins are intrinsically disordered, block copolymeric proteins that allow cephalopods to own one of the most strategic and sophisticated animal coloration systems in nature, producing iridescent camouflage and signaling. Reflectins are composed of highly conserved domains enriched in Met, Arg and Tyr, with barely 1% of aliphatic residues (reflectin motifs) and distributed within cationic linkers. This single very highly-conserved N-terminal motif is found in most reflectin proteins and differs in its N-terminal half from the repeating motif [105]. The N-terminal half of the protein likely plays a fundamental role mostly in determining the assembly and biophotonic-controlling properties of the proteins. What drives this dynamic iridescence in cephalopods is the reversible assembly of reflectin proteins. In particular, tunability is correlated with the presence of one specific reflectin sequence. However, reflectin subtypes possess different roles with different tissue-specificity and subcellular spatial distributions, as well as a differential phosphorylation and dephosphorylation response to activation by acetylcholine. Specifically, reflectins drive a Gibbs–Donnan-mediated efflux of water across the lamellar cell membrane, rapidly and reversibly dehydrating the lamellae to thus change both the periodicity and the refractive index contrast of the multilayer reflector, accounting for the tunability of the color and intensity of reflectance [106]. In vitro studies have shown that photonic changes cause reflectin phosphorylation, and that neutralization of phosphorylated reflectins drives their condensation to assemble hierarchically. Levenson et al. [107] explains: “the reflectins function as a signal-controlled molecular machine, regulating an osmotic motor that tunes the thickness, spacing, and refractive index of the tunable, membrane-bound Bragg lamellae in the iridocytes of the loliginid squids”. In several studies, recombinant reflectin-based thin films exhibited dynamic color changes by exposure to acetic acid vapor [108], application of water vapor pulses [109] or structural/assembly manipulation by spin coating [110]. Reflectins may also be chemically or genetically modified post-fabrication to control optical properties such as light scattering, UV reflectivity or broad reflectance in the near/short-wave IR regions [109,111].

### 3.3. Protein Based Metafluids

Parallel to the term metasurface, metafluid is a colloidal suspension of plasmonic metamolecules to accomplish unconventional bulk, specially exhibiting magnetic responses [3]. In contrast to planar metamaterials, metafluids can provide versatile solution processability, making them more effective in various applications [112]. Liquid materials such as liquid crystal, liquid metals and even water are applied to obtain a metafluidic metamaterial [113]. Electron beam lithography (EBL) or focused ion beam (FIB) have usually been used for modeling metamaterials but they have prohibitive costs [114]. In contrast to these techniques, nanoparticle self-assembly has the potential to produce high-performance three-dimensional geometries with relatively low capital investment. Self-assembly techniques also support colloidal processing, providing a platform for the synthesis of metafluids, as metafluids creation requires a high yield of assembled metamolecules [115].

Ligand binding triggers and modulates the biological function of many proteins. Ligand use offers a great variety of new synthetic methods for the creation of new nanostructures applied in the manufacture of nanodevices [116], including metafluids. These assembly methods take advantage of the architectures, nanometric dimensions and diversity of functional groups present on the surface of proteins and peptides. The specificity of protein-based chemistry has been used by Sheikholeslami et al. [115] to lead the random aggregation of colloidal silver nanoparticles in the synthesis of an isotropic metafluid on gram-scale (Figure 3b). The obtained metafluid exhibited a strong magnetic response at visible frequencies and a negative refractive index at modest fill factors. In their work, silver nanoparticles were functionalized with biotin terminated ligands and mixed in a high ionic strength buffer with streptavidin-coated polystyrene nanoparticles to form the core of metamolecule. Streptavidin is a tetrameric protein synthesized by the bacterium *Streptomyces avidinii* with high affinity for biotin. The streptavidin-biotin binding induces an increase in protein tightness and leads to a higher thermostability [117].

### 3.4. Protein Templated Metamaterials

Chiral metamaterials are of great interest because of the possibility to induce a negative index of refraction, resulting in unique optical functionalities that may be applied in novel types of optical devices as circular polarizers or chiro optical sensors [118]. Some secondary protein structures and peptides are chiral biomolecules known to assemble in various fibrillar structures adopting chiral morphologies including helical structures. Therefore, some proteins and peptides have been used as chiral nanoparticles assembly agents applied to nanoparticle architectures [119].

An interesting chiral protein structure is the amyloid fibril. These fibrils were identified as pathological entities in several neurodegenerative diseases, dialysis-related amyloidosis, and chronic inflammatory disorders [120]. Nevertheless, new studies have proposed that amyloid fibrils serve as functional protein assemblies to achieve a wide range of biological functions [121]. Amyloid fibrils are obtained when the native protein structure changes to an energetically favored non-native structure that resembles a nano-fiber and is rich in β-sheet secondary structure. The amyloid fibril is thermostable, stabilized by supramolecular noncovalent bonds, and by the richness in closely packed and highly ordered β-sheet patterns [122]. Various studies have shown that different molecular structures, morphologies and biochemical characteristics are present in amyloid fibrils associated with variants of the same disease or in different pathological states, due to the diversity of protein components that constitute them as well as the conditions of its microenvironment [123]. This great variety of molecular architectures that is established in the structure of amyloid fibers can be exploited and refined for the specific 3D accommodation of nanomaterials.

In this context, Leroux et al. [124] performed the synthesis of a metamaterial based on metallic nanoparticles attached to active sites on the surface of an amyloid template. In these experiments, bovine insulin was treated at pH 2 and 70 °C to promote its polymerization. Under these conditions, insulin amyloid fibrils consisted of 2 to 6 intertwining protofilaments and 4 to 6 nm of diameter and several micrometers of length. Then, they attached silver nanoparticles to active sites on the insulin fibrils by electroless plating. After blotting off excess liquid, the silver ions were reduced by addition of a reducing agent. The relevance of this report was the development of a strategy for the large-scale production of chiral metamaterials with tunable optical properties by the deposition of metal nanoparticles on protein amyloid fibrils. Several studies about the use of insulin fibers as template for the controlled synthesis of metal nanochains have been developed with different metals as gold [125] or platinum-palladium [126]. Others even conjugated the fibers with Alexa fluor dyes [127], demonstrating that these highly regular and steady structures are an effective method to construct bioconjugated metal nanomaterials.

Just as insulin under certain specific treatment conditions, apoferritin or β-lactoglobulin have been recently used as scaffolds to prepare different metal biomaterials thanks to the orderly and stable formation of amyloid fibers.

β-lactoglobulin is the major whey protein in the milk of most mammals with a cost-effective recovery process [128]. Recently, Suo et al. [129] incubated β-lactoglobulin at 90 °C and pH 2.0 to induce the formation of amyloid fibers, which were later used as structural scaffolds of gold nanoclusters. Gold nanoclusters consist of several to hundreds gold atoms organized in ultrasmall structures (diameter ≤ 2nm) with strong quantum confinement effects, and have shown wide applications in the fields of sensing, imaging, and drug delivery [130]. β-lactoglobulin fibers are polymers containing partially dissociated ionic groups in aqueous solution, with strong, long-range electrostatic interactions. Thanks to this, their surface will tightly bind with AuCl_4_^−^ ions present on gold nanoclusters. Combined, the optical characteristics of gold nanoclusters and the stabilizing and reducing capacity of β-lactoglobulin amyloid fibers produced materials with different color fluorescent emission at different pH, producing green emission at pH 1, blue emission at pH 8, and red emission at pH 12, under 37 °C. In the same study, the authors evaluated the fluorescence imaging in vivo, using A549 cells. The cells were incubated with the three luminescent gold nanoclusters templated on β-lactoglobulin fibers under controlled incubation and wash conditions. The luminescent gold nanoclusters templated on β-lactoglobulin can achieve multicolor at intracellular conditions, indicating a huge potential of application in fluorescent imaging [129].

Ferritins are iron-storage and detoxifying oligomeric proteins in most organisms and prevent the harmful accumulation of iron by collecting free iron in the form of ferrihydrite phosphate [131]. Not long ago, Jurado and Galvez [132] used apoferritin amyloid fibrils as bio-scaffolds for the preparation of different metal nanomaterials. They conjugated apoferritin fibrils with gold, silver and palladium nanoparticles, gold nanorods, gold nanospheres and magnetic iron oxide nanoparticles. All apoferritin bioconjugates were compared to its β-lactoglobulin bioconjugates contraparts as models. Additionally, they developed a second novel in situ method where apoferritin fibril precursors are used for the synthesis of the nanoparticles. In both strategies the nanomaterials were well absorbed and structured. These results demonstrated that apoferritin amyloid protein structures have a high affinity for a wide range of metals (Figure 3c). The gold nanostructures-apoferritin fibrils conjugate exhibited potential use as optical imaging enhancer due to plasmonic properties based on their absorption bands identified in the localized surface plasmon resonance. Amyloid fibrils biotemplated materials have high plasticity, structural stability and functional integrity, allowing its application in other metamaterial configurations such as hydrogels.

### 3.5. Protein Hydrogels

Several hydrogels have been made from fibrillar proteins such as silk, collagen, mini collagen 1, keratin and even with the intercalation of other proteins such as bone morphogenetic protein-2 or cellulose crystals. Most of these efforts are focused on tissue engineering for repair and regeneration, due to inherent biocompatibility and mechanical properties [133,134,135,136]. However, because of properties such as high transparency, and thermal resistance there is recent interest in the development of hydrogels for applications in photonic devices [101].

Amyloid-based hydrogels are attractive and have great potential application in the biomaterials field scaffolds for tissue engineering, drug delivery, nano devices, and food technology [1]. Biomacromolecules can build a hydrogel scaffold directly under variation of temperature, pH, ionic force, and salt concentration. These selected conditions facilitate self-assembly and the coupling reactions. For example, Tang et al. [137] converted amyloid fibrils into hydrogels by controlling the concentration of proteins/peptides and the temperature of the solutions. Elsewhere, hydrogel formation by lysozyme fibrillation was directed adding tris(2-carboxyethyl)phosphine (TCEP) at a molar ratio of 4:1 and with a heating and cooling combination of steps. First, heated at 60 °C inducing lysozyme unfolding, then cooled down to 25 °C. The result was a lysozyme hydrogel optically transparent at lower molar ratios of TCEP/lysozyme converted into turbid gels. The turbidity of the gels decreases with the increase in the concentrations of TCEP [138]. Heat at 60 °C with TCEP addition leads to significant perturbation in the structure of the lysozyme folded state due to reduction of all four disulphide bonds present in the lysozyme structure [139]. Lower concentrations of TCEP caused turbidity in hydrogels due to incorrect disulfide bond patterns, formed by misfolded and partially reduced lysozyme at that condition. Consequently, transparency depends on a post-traductional modification of the protein employed in this hydrogel fabrication strategy (Figure 3d).

The afore mentioned works destabilized sophisticated structures of native proteins by high temperatures and low pH to generate amyloid fibrils. However, they do not take advantage of the great structural diversity of proteins and the selective and specific interactions that can be established on the protein surface. Cross-linked polymeric networks of hydrogels can be modified to be stimuli-responsive components by the attachment of some proteins such as GFP or specific antibodies.

Kim et al. [140] developed the first method of preparing fluorescent silk fibroin solution in order to produce various fluorescent SF materials. Coding sequences of fluorescent proteins (EGFP, mKate2, EYFP) were genetically fused to the N-terminal and C-terminal domains of the silk fibroin H chain by molecular biology techniques. The generated vectors pBac-3xP3-DsRed2-FibH-EGFP, pBac-3xP3-DsRed2-FibH-mKate2, and pBac-3xP3-DsRed2-pFibH-EYFP contained the coding sequences for silk fibroin and for each of the different fluorescent proteins used in the study. These vectors and the helper vector pHA3PIG were injected into pre-blastoderm embryos at 2–8 h after oviposition. Injected embryos were allowed to develop at 25 °C in moist chambers. Fluorescent silk fibroin variants were expressed by these transgenic worms. The fluorescent cocoons were treated to remove sericin proteins to obtain a free fluorescent silk fibroin solution. To prevent intramolecular and intermolecular disulfide bonds formed between cysteine residues of proteins, the silk fibroin solution was added with DTT and LiBr, and incubated at 45 °C. Then, the solutions were dialyzed to obtain silk fibroin solutions with green (EGFP), red (mKate2) and yellow (EYFP) fluorescent colors, respectively. Upon confirming the fluorescent capability of these silk fibroin, they fabricated different forms of materials and applied them in tumor visualization tests by fluorescent SF solution conjugated to p53 antibody. Fluorescent-labeled p53 antibodies were used to detect p53 expression in HeLa cells derived from cervical cancer cells in vitro. Similar efforts were previously reported by the same research group with blue fluorescent protein (EBFP) and red fluorescent protein variant2 (DsRed2) [141,142].

Table 2 describes some works focused on the development of protein-based metamaterials, addressing the proteins involved, manufacturing strategies, and even their optical properties.

**Table 2 biosensors-12-00962-t002:** Examples of protein based structured material and their optical properties.

Protein	Configuration	Method	Properties	References
Insulin amyloid fibrils	Silver particles (5 nm) follow the helical structure of the biomolecule used as a supporting scaffold.	Amyloid fibrils induction by exposition of insulin to pH 2.0, 70 °C during 7.5 h. Silver NPs were attached to active sites on the insulin fibrils by electroless plating. AgNO_3_ solution was added to the insulin fibrils (70 mM) and left to incubate overnight.	Optical Chirality. Distribution of NPs deposited along biomolecules, without the use of heavy-metal staining. A general approach for aligning NPs into chains on functional surfaces for applications such as optical waveguides.	[124]
Insulin amyloid fibrils	Steady Pt-Pd NP chains templated on Insulin amyloid fibrils with an uniform diameter of about 10 nm and lengths up to several μm.	Amyloid fibrils induction by exposition of insulin to pH 1.6, 70 °C during 7.5 h. Site directed adsorption and subsequent reduction of PtCl_4_ and Na_2_PdCl_4_ during 2 h at room temperature, inducing the growth of Pt-Pd NPs in situ to enable them to align regularly into chains.	Optical Chirality and catalytic activity, CO oxidation and methanol oxidation.	[126]
Apoferritin (APO) and β-lactoglobulin protein amyloid fibrils	Au, Ag, Pd NPs, Au NRs, Au NSs and magnetic iron oxide NPs.	Amyloid fibrils induction by exposition of Apoferritin and β-lactoglobulin to pH 2.0, 90 °C during 24 h. To prepare bioconjugates, AuNSs or AuNRs solution was added to Apoferritin and β-lactoglobulin protein amyloid fibrils fibril solution at pH 8 and incubated for 24 h	Highly magnetized nanomaterials (magnetic anisotropy) due to the long-range dipole-dipole coupling and the MNPs alignment along amyloid fibrils.	[132]
Streptavidin-biotin system	Isotropic metafluid. Subwavelength structure of strongly coupled nonmagnetic NPs. An individual metamolecule is modeled as 32 Ag-NP (36 nm) surrounding the surface of streptavidin coated polystyrene NP (90 nm), with an average interparticle spacing of 3.8 nm.	First, citrate-capped Ag-NP (36 nm) were synthesized and functionalized with biotin-terminated PEG ligands. The biotin-functionalized Ag-NP were added to a solution of streptavidin coated polystyrene NP (90 nm).	Metafluid exhibited strong optical-frequency magnetism response at visible frequencies. The highly specific chemical recognition Streptavidin-biotin allows the Ag-NP closely and symmetrically pack around.	[115]
Bovine serum albumin (BSA) β-lactoglobulin from bovine milk (BLG) Conalbumin from chicken egg-white(CA) Recombinant human insulin Cytochrome c from bovine heart (Cyt)	Plasmonic Raspberry-like Core/Satellite Nanoclusters. Spherical Au or AgNPs with a hydrophilic protein shell (as satellites; 5–32 nm) onto large sized metal NPs (as cores; 45–100 nm).	AuNPs or AgNPs were synthesized by the citrate reduction-based seeded-growth method. Citrate-stabilized satellite NPs were coated with a different protein (BSA, BLG, CYTC, and CA) using a simple ligand exchange process at pH 9.0. The citrate-stabilized NP dispersions was added dropwise to protein/citrate solution under vigorous stirring during 24 h at room temperature.	Tuning of the optical/plasmonic properties by changing particle size, composition, or assemblies in a broad range of the visible spectrum.	[143]
Lysozyme	Hydrogel	Hydrogel formation by a combination of heating (60 °C) and cooling steps (25 °C) on a lysozyme/TCEP mix at a molar ratio of 4:1. The obtained lysozyme hydrogel was optically transparent. At lower molar ratios of TCEP/lysozyme converted into turbid gels.	Transparent or cloudy optical appearance depending on the concentration of lysozyme or TCEP as well as the effect of the redox, heating and cooling conditions applied on the hydrogel fabrication due to formation of misfolding lysozyme intermediates in the structural organization of the hydrogel.	[138]
Silk Fibroin	Periodic nanopatterned 2D lattices in pure silk fibroin protein films. (lattice spacing 300–700 nm, thick film ∼10 μm).	Silk nanostructures were created by nanoscale electron-beam writing of hard metal masks.	Periodic lattices in silk fibroin films feature sizes of hundreds of nanometers that exhibit different colors as a function of varying lattice spacing, angle of illumination and angle of collection.	[98]
Silk fabrics	Spherical AuNP in situ synthetized on the surface of silk fabrics (AuNP diameter 28.3–134.7 nm).	AuNP were synthesized in situ. Solutions containing white silk fabric samples and different concentrations of HAuCl_4_ (0.1–0.6 mM, 50 mL) were shaken for 30 min at room temperature before heating. Subsequently, the solutions were heated at 85 °C for 60 min in a shaking water bath.	AuNP on the surface of fibers leads to yellow, red or brown colors of the treated silk and give rise to enhanced electromagnetic fields for amplification of optical signals.	[99]
Silk fibroin	Silk fibroin paper with fibers of 20–120 nm in diameter. Paper thickness varied from 75 μm to 55 μm according to the process applied.	Silk from *Bombyx mori* was treated to eliminate the sericin protein, degummed fibroin was ground and subsequently homogenized (1, 9, 15, and 30 times). The homogenization system allows fibrillation. Each silk fibroin “pulp paper” was filtrated and dried.	Optically transparent silk fibroin nanofiber paper. Paper thickness, transmittance at 600 nm, transparency varied according to the number of homogenization passes (1 pass to 30 passes).	[104]
Recombinant reflectins proteins SoRef2, SoRef1, and SoRef8	Spin-coated reflectin films. SoRef2 film exhibited a relatively smooth surface with a roughness ∼11.6 nm (without imidazol) and ∼65.4 nm (with imidazol).	Spin coating of reflectin films was performed in a cleanroom. Approximately 250 μL of each protein (SoRef2 at 330 mg/mL, SoRef1 at 416 mg/mL, or SoRef8 at 386 mg/mL) was pipetted onto the center of a pre-cleaned glass slide or polished silicon wafer placed on a spin coater, which was operated at 2000 rpm for 50 s to generate films with different thicknesses.	Reflectin films formed by higher-order assembled structures exhibited dynamic color changes from colorless to white and blue, regardless of film thickness and the type of reflectin protein.	[110]

**Figure 3 biosensors-12-00962-f003:**
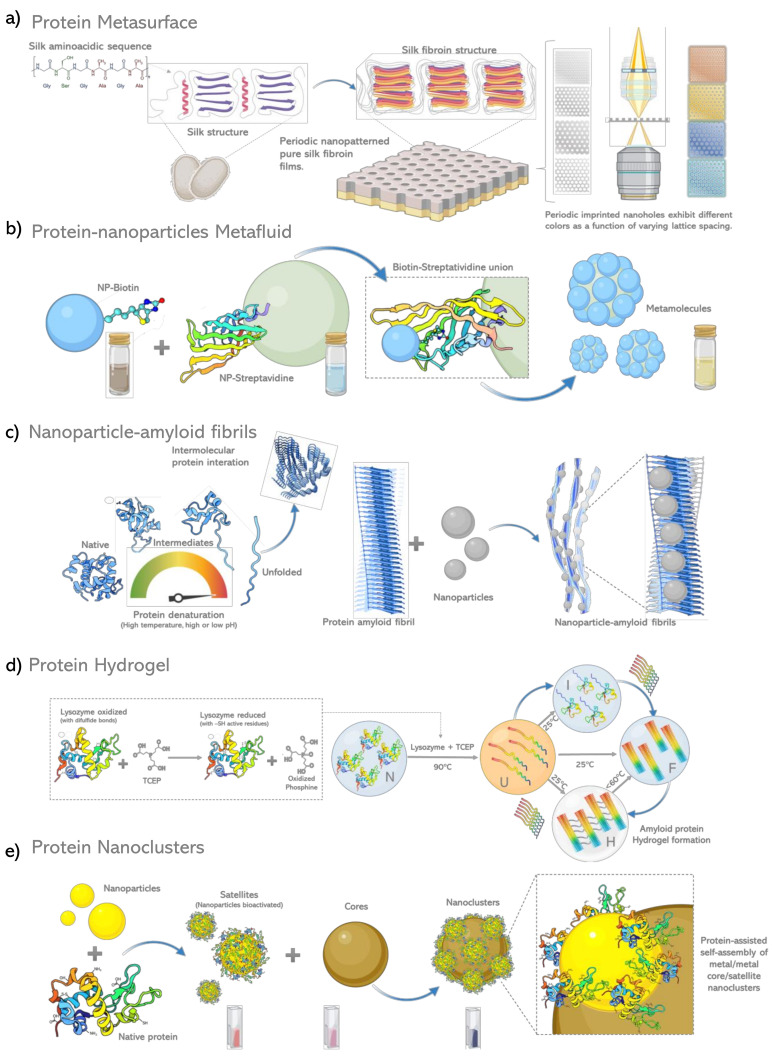
**Schematic diagram of protein based metamaterials.** (**a**) Protein films of silk fibroin nanopatterned 2D lattices that exhibit different colors as a function of varying lattice spacing [98], (**b**) Protein affinity interactions direct the self-assembly of metamolecules integrated by NPs tightly packed around a single dielectric core [115], (**c**) Protein amyloid fibrils induction by chemical and thermal denaturation, site directed adsorption and subsequent reduction of precursor salt inducing the growth of NP´s nanoparticles aligned regularly into amyloids fibrils [132], (**d**) Formation of protein amyloid based hydrogel by reduction, concentration variation and thermal denaturation process [138], (**e**) Coated Protein based self-assembly of metal/metal plasmonic core/satellite nanoclusters [143].

### 3.6. Strategies for Protein Production

Proteins addressed in this paper such as silk, albumin, amyloid fibrils, lysozyme, albumin, reflectins, or FPs (Fluorescent proteins) have unique intrinsic attributes for optical metamaterials design. These proteins offer advantages such as tunability, self assembly, biocompatibility, reduction of costs and environmental impact. Cost reduction and environmental impact depend on the recovery and production processes followed, even more so if we consider that many of these proteins can be extracted from industrial food wastes. Processing of natural resources in the food, beverage, cosmetic, and pharmaceutical industry generates a significant amount of byproducts and waste. They have been used as a source of bioactive compounds with high value including dietary fibers, phenols, lipids, hydrocolloids, and proteins [144,145]. For example, the demand for egg and egg products has increased in the world, leading to the generation of large amounts of wastes, in particular eggshell. Proteins such as lysozyme, conalbumin, avidin, and ovotransferrin are mainly present in egg white, but also in the eggshell matrix [146]. Egg proteins have been applied in many fields due to their foaming, coagulative, emulsifying, antimicrobial and structural intrinsic properties [147]. In the area of optical metamaterials, lysozyme has been used in hydrogel fabrication with transparent or cloudy optical appearance depending on lysozyme concentration [138]. Conalbumin has been applied for the construction of nanoclusters with different optical/plasmonic properties by changing particle size, composition, or assemblies (Figure 3e) [143].

The proteins derived from the dairy industry are additional proteins available based on food infrastructure already in place, such as whey from dairy liquid byproducts. In 2014, world production of whey protein was 240 million metric tons with an expected increase rate of 3.5% annually. Another byproduct is lactalbumin from bovine serum milk, one of the major serum proteins [148]. Whey proteins have been widely studied and applied, particularly in the food industry, due to their nutritional value. However, recently, many studies reported their use as building blocks of nanoparticles, nanofibers and hydrogels [149].

Conventional protein separation and purification from food waste include chemical or thermic precipitation, preparative liquid chromatography, filtration techniques (microfiltration, ultrafiltration, reverse osmosis), and spray-drying, among others. These operational units are organized and conjugated to find a more cost-effective strategy to separate and purify proteins efficiently, while maintaining their function-activity [150]. Nevertheless, in order to increase recovery of high value components from food waste, it is necessary to develop novel extraction strategies and improve the overall eco-sustainability of the process [151]. Food waste proteins could be recovered and applied in the development of sustainable products in relevant industries to create innovative solutions.

On the other hand, standardized industrial processes are used exclusively to recover and purify proteins with high potential for the development of metamaterials. For example, keratin and silk are harvested from the textile industrial production of wool and cocoons, respectively. Silk is a hypoallergenic biomaterial with multiple properties that make it attractive to a wide range of applications, from material engineering to medical device development. Nevertheless, its applications are widely curtailed by the practical challenges of production scalability, due to the cannibalistic and territorial nature of spider farming [152,153,154]. Although protein production by recombinant technologies is an alternative, it presents operational limitations or low recovery yields.

The biggest advantage of recombinant technology is that almost any protein can be expressed in living cells. By genetic sequence insertion, the cells will produce the encoded residues that define the protein structure and physical characteristics.

Multiple strategies for recombinant silk production have been proposed [152,153,154,155]. At first, the most viable production used transgenic silkworm (*Bombyx mori*) [152,154]. The insertion of the gene MaSp1 to *B. mori* showed the viability of large-scale production of silk in the animal [154]. Nevertheless, the use of this model limits the production to silk fibers and does not allow for isolated protein production, limiting its engineering possibilities. In response, research groups proposed the production of silk in bacterial, yeast, animal and plant systems. Of special interest is the production of recombinant silk in *Escherichia coli.*. Edlund et al. [152] forecasted that *E. coli* silk production may lower the costs (as low as $23 USD per kg) and reduce the environmental impact (with a minimal carbon footprint of 55 kg CO_2_ eq. per kg of silk). Nevertheless, recombinant silk production in *E. coli* faces more complex challenges than the standard recombinant protein production, as silk protein is highly hydrophobic and tends to form aggregates, limiting its productivity and the amount of correctly folded protein obtained [155]. To overcome this challenge, Abelein et al. [156] mutated the N terminal of the spider silk protein subunits by inverting its general charge resulting in higher recombinant production yield in *E. coli*. This shows protein engineering potential for more efficient and less resource-intensive production of recombinant silks.

The major advantage of biological systems to produce proteins of interest is the possibility to manipulate and optimize their physical and chemical characteristics through mutagenesis, enabling genetic engineered processes. As an example, GFP and reflectins have been tailored to perform specialized tasks [111,141,142]. These modifications allowed new protein properties, which lead to new optical properties of the materials where they were used.

Despite advances in protein production through biological expression in prokaryotic or eukaryotic systems, chemical synthesis has emerged as an attractive alternative to introduce homogeneous and site-selective specific modifications in a protein of interest [157]. Solid phase peptide synthesis (SPPS) is the most efficient technique for polypeptide chemical preparation. SPPS involves numerous repetitive coupling reaction steps on a target nascent peptide immobilized to a solid support by a linker [158,159]. Although SPPS allows efficient synthesis of small proteins, the use of highly toxic reagents and the need for special equipment limit the applicability of this approach to research scale only [160,161]. Protein chemical synthesis needs considerable improvement to produce larger proteins by simpler chemical methods. A brief diagram that encompasses various sources and production methods of proteins that could be used for the development of metamaterials is found in Figure 4.

## 4. DNA/RNA Based Metamaterials

The incorporation of soft materials (such as liquid crystals, fluids, biomaterials, and polymers) in the design of metamaterials offers a versatile, tunable, easy production. This results in materials with biocompatible and adaptable characteristics that have various applications [3]. The nanoscale engineering of DNA facilitates the self-assembly of complex structures, and it is used as a building block of chiral metamaterials [16].

DNA possesses exceptional properties such as biocompatibility, molecular recognition ability, programmability and nanoscale controllability derived from its biological function and its structural characteristics as carrier of genetic information [162]. DNA is a powerful tool for the rational assembly of plasmonic nanoparticles, superlattices and hydrogels that exhibit metamaterial properties (view Table 3 for examples mentioned in this review). During the construction of DNA based metamaterials, DNA can play a role as a substrate in the case of biochemical reactions involving enzymes. It can also act as a linker that links DNA interfaces with functional moiety, such as proteins, nanoparticles, or DNA itself [163]. Interactions occur between the purines and phosphate moieties of DNA and selected metal species (e.g., Au, Ag, Cu, Pb), semiconductors (e.g., Si, Se, Te) and polymers. In particular, nitrogen and oxygen atoms play a dominant role in metal/phosphate coordination by electrostatic interactions [164]. The great interaction versatility of genetic material allows for the creation of hybrid nanostructures with sequence-dependent control.

Traditionally, nanostructures are fabricated through top-down methods such as lithography, where nanoscale features are “carved” from the bulk materials [165]. The alternative from the bottom-up, uses small building blocks such as biomolecules and colloidal inorganic/organic nanoparticles that are guided to assemble into desired architectures over a large scale. DNA has an important role in bottom-up methods [166]. The implementation of DNA as building material has followed two strategies: materials composed of DNA and materials composed with DNA [167]. The first strategy, intentionally placed base pair interactions, bend or fold DNA strands into well defined 2D and 3D structures. DNA origami belongs to this category and it employs DNA strands as templates to compose into sophisticated structures [168]. DNA origami design consists of converting the target shape into a geometrical model equivalent with DNA double helices, and then filling the shape with single-stranded DNA derived from the bacteriophage M13 [169]. DNA origami technique has been widely used in drug delivery, artificial nanopores, single molecule studies, nanometrology, macroscopy standards, biosensing, imaging and nanophotonics, among others [170]. In contrast to lithography methods, DNA origami provides an alternative bottom-up fabrication route for plasmonic materials with controlled optical responses in the visible spectral range and not restricted by resolution or three dimensional architecture [171]. The second strategy consists of crystal engineering with DNA, where the properties of nanoparticles and DNA strands are combined to manufacture various materials. For example, DNA-coated colloids that self-assemble into an incredible diversity of crystal structures [172].

Most biomolecular nanotechnology applications use DNA or protein-based nanoparticles to create materials with optical properties. Circular dichroism in the UV range can be transferred into the plasmonic frequency domain when metal surfaces and chiral biomolecules are in close proximity [173]. Diversity of nanostructures or material hybridized with DNA have been characterized with an increase of CD signal in the visible region. For example, AuNR with folded G-quadruplex, ssDNAs attached on the surface of Au/Ag nanocubes, cyanine dyes positioned on NP surface through DNA-templated molecular self-assembly [174,175,176]. Additionally, the fabrication of a light-driven plasmonic nanosystem, able to translate the molecular motion into reversible chiroptical was fabricated by combining azobenzene with DNA molecules [17].

Moreover, with advances in genomics, proteomics and the new understanding of RNA, nucleic acid-based nanotechnology has been widely studied [177]. Additionally, nucleobases (the small molecules that constitute DNA and RNA), are useful as moieties for the ecological synthesis of various organic and organic-inorganic materials [178,179,180]. In this review we will show some relevant examples of structures and materials with optical properties and metamaterial characteristics made with DNA and RNA. In addition, we will explore the main methodologies for producing each type of biomolecule for its use in the engineering of novel metamaterial structures.

### 4.1. DNA

#### 4.1.1. Spherical Nucleic Acids (SNAs)

The initial research with DNA and nanoparticles started in 1996 with Mirkin, they associated free DNA-modified particles suspended in solution with dispersed gold nanoparticles. The addition of a complementary DNA strand brought the particles closer together and the color turned purple within minutes. This work gave rise to a new class of DNA/nanoparticle hybrid materials and constructs, demonstrating that DNA could be used to assemble gold nanoparticles into periodic structures [181]. Later, the controlled aggregation of oligonucleotides on the surface of organic/inorganic nanoparticles or nanomaterials, under varying parameters and using diverse materials, received the name of Spherical Nucleic Acids (SNAs) [182,183]. These DNA-grafted nanoparticles have also been called “programmable atom-equivalents” because they form three-dimensional crystals, but unlike atoms, the particles themselves carry information that can be used to program its crystal structure [184]. When SNAs adopt a regular periodic pattern arrangement in either 2 or 3 dimensions, they form complex structures, such as diverse lattice configurations. Nanoparticle-based lattice materials allow higher stability control by modifying nanoparticle shape or size, or by the presence of molecular layers on its surface [185]. The predictability and programmability of DNA makes it an excellent tool for the precise assembly of plasmonic nanoparticles with tunable nearest-neighbor distances and symmetries in one, two or three-dimensional lattices. Crystallographic symmetry can be adjusted by linker lengths, linker sequences, NP diameters and molar ratios [182].

Typically, the core of a SNA is made up of inorganic materials such as metals (Au and Ag), metal oxides, quantum dots and semiconductors. Noble metal nanoparticles have been widely applied as inorganic core materials, being gold nanoparticles (AuNPs) the most representative. AuNPs are optical building blocks that have localized surface plasmon resonance (LSPR) excitation, which is sensitive to NP size, shape, composition, dielectric environment, and proximity between plasmonic NPs [186]. AuNPs can scatter light to produce diffracted waves that propagate in the plane of the array. This leads to a drastic narrowing of plasmon resonances to 1–2 nm in spectral width [187]. Ordered arrays of plasmonic AuNPs exhibit a variety of important properties for the development of metamaterials, such as the ability to guide light around sharp corners, a broadband optical response, Fano resonances, and a negative index [188]. The change in distance between individual AuNPs allows for the configuration of their optical properties. AuNPs are widely employed due to their ease of functionalization with biologically active organic molecules or atoms [189]. AuNPs’ surface can directly conjugate and interact with diverse molecules containing such as proteins, drugs, antibodies, enzymes, fluorescent dyes and nucleic acids (DNA or RNA).

Although SNAs most commonly involve inorganic nanoparticles in their preparation, this raises long-term toxicity concerns due to the tendency to accumulate in organs [190]. Therefore, biodegradable and biocompatible organic nanostructures have emerged, such as liposomes, proteins and poly lactic-co-glycolic acid (PLGA) [183].

#### 4.1.2. Nucleic Acid Biodots

Nanometer-sized semiconductor crystals or ’quantum dots’ (QDs) have a wide range of applications due to their unique fluorescent and electronic characteristics, mainly simultaneous excitation in multiple color spectra, size-dependent light radiation, long-term photostability, and high signal brightness [191]. QDs have applications in different areas such as optoelectronic devices, biological imaging, solar energy devices, biomedicine, among others [192]. Commonly used QDs are either prepared by using Cd, Pb, Cu and other elements with high environmental toxicity, or at high temperatures (more than 100 °C) with the use of toxic organic solvents, which limits their practical application. The synthesis of QDs has evolved toward the use of safer alternatives, such as Ag(I), Cu(I), and carbon dots [193,194,195]. The process of functionalizing QDs is not only complex and time-consuming, but also unsatisfactory in stability. Therefore, directly using ligand molecules with low toxicity targeting recognition groups as templates for QDs sensor, that results in high specificity and stability has become the main strategy to solve those problems. Nucleic acid structure has an interesting effect on the fluorescent properties of nanoparticles (biodots) [196]. Individual nucleotides (AMP and GMP) are more suitable for biodot synthesis compared to DNA and RNA macromolecules. Biodots prepared from AMP and GMP nucleotides showed superior fluorescence properties and quantum yields compared to the products from other nucleic acids and other natural compounds, such as amino acids and saccharides. Although the quantum yield of biodots is lower than that of classical alloyed semiconductor quantum dots, biodots have additional advantages that favor its use. Biodots have no toxicity, can be prepared from renewable resources, and their fluorescence is stable in a broad range of pHs and cations [197]. Biodots can be applied in biomedical applications (bioimaging, sensing, drug delivery, and the analysis of body fluids) and can also be used for detection of environmental pollution with heavy metals, due to the recognition properties of nucleic acids toward metal ions [198].

Other variants of hybrid materials with QDs have been reported, such as the pH responsive carbon dot CD-DNA hybrid hydrogel for targeted and sustained release of drug molecules. The conjugation of a functionalized surface of CDs with Cytosine-rich ssDNA formed a stable network in a hydrogel. The intense fluorescence of the CDs was retained in the hydrogel that may be potentially tracked through imaging to correlate the efficacy and level of drug loading [194]. QDs represent metamaterials with huge opportunities for different optical and catalytic functionalities that could provide novel tools for imaging and probing chemical and biological interactions at a molecular level, enabling characterization, functionalization and other real-life applications.

There have been many studies in which DNA origami has been used as a template to position a QD. Colloidal quantum dots have electronic and optical properties that, depending on the size of light wavelengths that they can absorb and emit, can be used for optical detection. DNA offers a high degree of bonding specificity, allowing widely different bond energies to be associated with different quantum dot pairings.

#### 4.1.3. DNA Hydrogel

DNA hydrogels are composed of complementary hybridized DNA molecules to form a highly structured network that expands upon hydration in an aqueous environment. DNA acts as the backbone of the network through complementary nucleotide binding, ligation, polymerization, and crosslinking of strands. This can be achieved by either non covalent or covalent bonds. DNA hydrogels are polymeric biomaterials with sufficient stability, biocompatibility, biodegradability and a variety of applications. These types of gels can be made by chemical or physical bonding of DNA molecules [199]. There are two categories, hybrid and pure hydrogels; the first subtype is assembled through the binding of nucleic acids to synthetic or natural polymers. Pure hydrogels are composed exclusively of DNA molecules and bind through, among others, Watson-Crick base-pair interactions, enzymatic ligation, and enzymatic polymerization [200]. Some of their most recent applications are their use in capillary assays for the optical detection of short cDNA from the SARS-CoV-2 virus [201], as potassium ion monitoring sensors with potential for in situ applications [202], such as hydrogel sensors integrated with quantum dot-linked transcription factors for quantitative detection of progesterone [203], among others.

Many applications of smart and functionalized DNA-hydrogel for the construction of biosensors to detect different analytes have been developed [113]. Currently, hybrid DNA hydrogels are primarily developed for fluorescence imaging by combining nanomaterials or biomolecules with unique functions. Bioimaging technology of DNA hydrogels allows biological structure and function research [88]. For example, a smart and versatile Au-DNA hydrogel (AuDH) system enables the accurate and sensitive differentiation of cancer cells by imaging intracellular multiplex miRNAs even [204]. Hydrogels can work not only as a cellular scaffold but also as a bidirectional optical communication channel for encapsulated cells. Interestingly, PEG-based hydrogels have in vivo biomedical applications of cell-containing optical waveguides [205].

#### 4.1.4. DNA Lattices

Configuration of DNA-modified gold nanoparticles can be programmed by the choice of particle type, DNA-pattern and sequence, in order to form optically active superlattices that can exhibit almost any color across the visible spectrum [18]. DNA nanostructures are suitable scaffolds for organizing nanoparticles due to the response to local environmental changes (Ionic strength, dielectric constant and so on) in a reversible and controllable fashion, resulting in lattice contraction or expansion driven by the conformation change of DNA. Additionally, DNA provides numerous choices of orthogonal motifs (hairpin,G-quadruplex, triplex, DNAzymes and aptamers) that can be exploited for a global phase transition. For example, the selective and reversible folding or unfolding of hairpins, can be used to control several determinant factors of particle interaction enabling lattice transitions between two or more states [206].

Multiple nanostructures have been developed, such as one-dimensional nanowires [207], two-dimensional (2D) nanolattice [208], and three-dimensional (3D) crystalline lattices [185]. More than 70 unique lattices have been synthesized by changing the characteristics of the DNA shell and particle core and some of the resulting structures have no natural equivalent [166].

DNA is an ideal linker to construct highly ordered nanoparticle assemblies in solid and solution phases. To form plasmonic superlattice sheets, the interparticle spacing can be precisely tuned by changing either the DNA ligand length or ionic strength. These superlattices can be patterned on a great variety of substrates (e.g., copper, carbon, silicon nitride), which will further facilitate any potential device integration [209].

#### 4.1.5. 1D DNA Nanostructures

##### (A) DNA Nanotubes

DNA can be used to form nanotubes using the methods of either tiled DNA motifs or DNA origami. DNA nanotubes are an interesting material whose chemical and structural properties allow their functionalization with other nanoparticles or biomolecules, such as proteins, and make possible their application in biology and engineering (Figure 5a). Additionally, DNA nanotubes can be used as biodegradable and biocompatible drug nanocarriers [210]. The size-selective encapsulation of gold nanoparticles (AuNPs) was achieved with triangular DNA nanotubes with longitudinal variation. With the selective release of the molecular cargo, these nanotubes have the potential to be used for the optical modulation of conductivity and guiding light on the nanometre scale below its diffraction limit [211]. Different strategies can be used to fabricate DNA nanotubes, for example, it was reported that DNA double-crossover tiles can be activated by an upstream DNA catalyst network to polymerize and form a DNA nanotube structurally analogous to biological microtubules [212]. Furthermore, the DNA motif of the six-helix bundle (6HB) and their variants were used to fabricate nanotubes as 1D self-assembled ordered structures that can be extended into two and three dimensions [213]. DNA origami nanotubes with quasi-one-dimensional characteristics provide a rigid structure for gold nanoparticle attachment in solution. Coupling of AuNPs on DNA origami structures can alter the shape of the DNA origami-AuNP conjugates. For that reason, several methodologies perform the immobilization and prealigning of DNA origami on nanorippled Si/SiO_2_ surfaces before nanoparticle assembly [207]. Nanotubes of more complex structure have been built using some DNA tiles or with origami methods as will be described in a later section.

##### (B) DNA Gold Nanorods Nanostructure

Gold nanorods (GNR or AuNRs) are a kind of plasmonic nanoparticle that shows polarization-dependent optical and electronic responses because of its anisotropic shape [214]. Nanorods are made of precious metals (gold, silver, platinum), that have a great biocompatibility, simple synthesis, and easy surface modification (Figure 5b). AuNRs have two plasmonic peaks, the first around 520 nm and the second at 800 nm. The latter one can be tuned by different synthesis conditions, which have been widely utilized for its application in biological imaging, detection and photothermal therapy [215]. Additionally, stimuli-responsive nanocarriers have been fabricated with gold nanorods in combination with cyclic ternary aptamers that carry chemotherapy drugs. In this case, DNA aptamers have an specific structure that binds specifically to the surface of malignant-type cells and, after laser irradiation, strong photothermal and drug delivery, a synergistic anti-cancer effect is observed [216]. On the other hand, another recent application is in the production of thin films of DNA and CTMA embedded with gold nanorods that were characterized by measurements of photoluminescence, X-ray photoelectron spectroscopy and ultraviolet-visible (UV-Vis) spectroscopy [217]. These films had potential applications for filters, blocking layers and biosensors, among others.

##### (C) DNA Nanowires

The creation of nanoscopic wires, 1000 times thinner than human hair, enables the creation of new technologies, such as the miniaturization in electronics [218]. Semiconductor nanowires have typical cross-sectional dimensions that can be tuned from 1–100 nm and lengths spanning from hundreds of nanometers to millimeters. The method of growth for the synthesis of nanowires has been explored with oxides, III-V compounds, elemental semiconductors, and metal nanoparticles (Figure 5c). Metallization processes have been attempted in order to transform DNA into electrically conducting wires by means of silver, palladium, nickel, and gold [219,220,221]. Superconducting materials have been used to increase conductivity of nanowires by the use of carbon nanotubes and DNA templates. Many nanoscale axial and radial nanowire heterostructures have also been designed and researched for their novel photonic and electronic properties. The assembly of diodes and active bipolar transistors has also been possible through nanowires.

##### (D) DNA-CTMA Nanofibers

The complex of DNA with a cationic surfactant has been used in optoelectronic applications because of its unique material properties, such as solubility in organic solvents, thermostable at temperatures as high as 180 °C, flexibility and its ability to form transparent films, membranes and fibers [222,223]. The most used surfantanc is Cetyltrimethylammonium, CTMA. CTMA bound to the backbone of purified DNA, has higher structural resilience due to its long alkyl chains of CTMA and it is not water soluble due to its hydrophobic nature [224]. This kind of material has potential for optical memory matrices, field-effect transistors, photochromic switches, electro-optic waveguides, and as electron-blocking layers for light-emitting diodes, LEDs [223,225]. Solid-state DNA based nanofibers (Figure 5d) can be an efficient matrix for FRET (fluorescence resonance energy transfer). Salmon DNA-CTMA nanofibers complexed with a donor-acceptor pair of fluorophores generate a white luminescent material [225]. The geometry of the electrospun fiber affects the emission intensity, while DNA facilitates emission of white light by imposing a spatial organization and a specific binding environment to the dye molecules. Interestingly, this material favors the conversion of UV light into white light, as was demonstrated by coating white-light-emitting DNAbased nanofibers onto 400-nm-emitting solid-state LEDs [225]. Dye-doped fibers of DNA-CTMA can greatly enhance fluorescent efficiency and show visible and near-infrared light waveguiding properties [226]. Light scattering and confinement in the fibers together with potential dye stabilization result in a significant reduction of non-radiative decay channels for excitation, ideal characteristics for a laser application [227]. Lasers based on biological materials are attracting an increasing interest for their use in integrated and transient photonics. Most common dyes are Rhodamine 6G (R6G) and hemicyanine because they show improved fluorescence in DNA. Typically DNA-CTMA is processed into thin films through ink-jet printing, dip-coating or spin-coating techniques. However, it involves a high loss of material, a considerable inconvenience due to the high cost of good quality DNA [228,229].

##### (E) DNA-Nanoparticles Chains

The use of PCR has been extended to solid surfaces such as nylon, glass, microtiter wells, and inorganic NPs. The intrinsic programmability, structural plasticity, and coordination interactions with NPs makes DNA a powerful molecular tool for large scale NP assembly (Figure 5e). Zhao et al. [230] used PCR to fabricate plasmonic NP chains of different sizes, denoted heterochains. The NPs were connected by DNA oligomers, alternating in a sequence big-small-big-small, spanning lengths from 40 to 300 nm. The continuous growth of heterochains during the PCR process was made possible by a tight control over the number of primers per gold NP, limited to only two per each big or small NP. The properties of the assembled superstructures prediction depended not only on the size, shape, and composition of the NPs, but also on the surface orientation and even spatial addressability. Finding that these DNA-NP chains displayed a strong plasmonic chirality at 500–600 nm, which was nonmonotonic in respect to the number of cycles. Au NP heterochains assembled in the course of 2–20 cycles provided an effective model to explore the role of the structure in the NP heterochains by measuring the surface-enhanced Raman scattering (SERS) intensity. The NP heterochains were found to greatly enhance the SERS signal of 4-ATP due to the interaction between Au NPs in the assembled chains, whilst the chiral response initially increased and then decreased with the number of PCR cycles. This demonstrated the potential of using such structures as efficient SERS-active substrates for sensing applications in DNA detection [230], which can be used for biosensing, plasmonic and electronic applications.

#### 4.1.6. 2D DNA Nanostructures

The fabrication of 2D structures involves mainly simple hybridization, crossover and origami techniques. In general, there are two main strategies for creating 2D DNA structures: short DNA strand-associated DNA tile assembly (also called static DNA nanostructures) and long DNA strands-associated DNA origami assembly [231]. DNA tiles are a set of artificial DNA nanostructures composed of several short single-stranded DNA (ssDNA) molecules and they use rational design of sequences with sticky ends and hierarchical assembly to form larger DNA structures [232]. The first 2D structures were based on DNA hybridization and multiple crossover junctions to impart sufficient rigidity to achieve directional interactions. DNA tiles with branches, such as the combination between sticky-end cohesion and branched DNA junctions, and the double-crossover (DX) molecules, are some examples of designs for the construction of 2D nanostructures or crystals. Furthermore, multiple DNA tiles have been created, for example, triple-crossovers (TX), Holliday junctions (HJ), paranemic crossovers (PX), JX2, as well as four-helix, eight-helix, and twelve-helix planar tiles; three helix and six helix bundles; parallelogram DNA junctions; cross-shaped DNA tiles; ssDNA tiles (also called DNA bricks); and triangular and three-point star motifs [14]. Regarding DNA bricks, each ssDNA has a unique sequence and acts as a molecular brick to interact with other bricks, forming 2D or 3D objects [233]. DNA brick designs can be used as modules, a set of DNA strands can serve as a 2D or 3D canvas, enabling the construction of a large number of arbitrary DNA objects. This method enables the build of a library of DNA objects with arbitrary sizes and shapes in a high-throughput fashion. Both DNA tiles, DNA bricks can be induced to form infinite-size crystal structures by bridging the head and tail bricks. Such 2D arrays are also useful as programmable scaffolds for the organization of nanoparticles and biomolecules. However, DNA origami platforms have been further exploited for the development of optical applications. Below we will describe some relevant examples of the progress with 2D structures.

DNA origami can be designed to self-assemble into different 2D geometric patterns, which in turn generate assorted DNA-framed nanoparticle arrays. Apart from being a nanoparticle carrier, surface anchored DNA origami can also act as a template for seeded metal growth or may also serve as a hard mask for surface patterning [234,235]. The application of a DNA-inorganic hybrid structure to achieve deep etching of a Si wafer for antireflection applications and the use of bare DNA origami as a mass in silica etching are examples that demonstrate the great potential of DNA origami in lithographical applications [236,237]. Additionally, the BLIN (biotemplated lithography of inorganic nanostructures) technique with DNA origami may enable complex nanopatterns for various optical applications [235]. Due to the flexibility and addressability of DNA nanostructures and the commercial availability of fluorescent labeled nucleic acids, the DNA origami approach has been used to produce nanoscopic calibration standards for stochastic reconstruction microscopy (STORM) and photoactivated localization microscopy (PALM) [238]. DNA origami based imaging has made great progress with the construction of novel fluorescent probes, termed metafluorophores and the development of the method qPAINT (quantitative Points Accumulation In Nanoscale Topography). Optical metafluorophore with tunable brightness was engineered using a 2D rectangular DNA origami [239]. This structure was functionalized to display multiple organic fluorophores in a compact sub-100-nm architecture. The precise spatial control over number, spacing, and arrangement of fluorophores on the nanostructures prevent the self-quenching and Förster resonance energy transfer (FRET) effect between dye molecules. On the other hand, qPAINT enables robust counting by analyzing the predictable binding kinetics of dye-labeled DNA probes [240]. This technique showed great precision counting strands from DNA nanostructures, proteins, and probes on mRNA targets in fixed cells [240,241]. 2D DNA origami structures mimicking the expected distribution and the number of molecules functioned for in silico and in vitro optimizations. Another important application of DNA-PAINT, is its use to obtain nanometrological traceability of nanoruler distances [242]. These DNA origami nanorulers have been used to evaluate the performance of TIRF microscopes and have the potential to become prime examples of self-assembled reference structures and traceable ubiquitous standards of length measurements [243].

Interestingly, the assembly of AuNRs on the surface of a DNA-origami structure (D-AuNR) has been used to develop applications in medical treatments, such as optoacoustic imaging (OAI), a molecular imaging technique that brings significant promise to enhance the depth of imaging penetration as well as spatial resolution, maintaining high contrast of optical imaging. D-AuNRs responded to NIR irradiation (photothermal therapy) and effectively inhibited tumor regrowth and prolonged the survival of diseased mice [244].

The precise nanoscale assembly of DNA origami has also allowed the creation of Optical nanoantennas by attaching one or two nanoparticles (Au or Ag) to DNA origami structures with available docking sites for a single fluorescent dye next to one NP or in the gap between two NPs [245,246]. These nanoantennas provide the possibility of increased fluorescence by plasmonic effects in the near-field of metal nanostructures more than 100 times the signal of the fluorophore [245]. Fluorescence enhancement in DNA origami nanoantennas is especially relevant for signal amplification in molecular diagnostic assays, such as the single-molecule detection of DNA specific to antibiotic-resistant bacteria on a portable smartphone microscope [247].

Moreover, nanostructure-paint through DNA allows the fabrication of planar optical metasurfaces, arrays of microscopic light scatterers, whose overall optical properties emerge from microscale structures, as it can be observed in the structural color of a butterfly wing [248,249]. Due to the excellent control over light, metasurfaces are typically envisioned for use in large-area applications such as windows and solar cells, functional coatings, holograms, wavefront shaping, and structural color printing [3,250]. Artificial metasurfaces can be engineered with clusters of metal nanoparticles, graphene ribbons or nanodisks [248]. Some methods of metasurface fabrication rely upon top-down techniques that are unsuitable for mass production, so advances in nanoimprinting point to it being a promising option. DNA origami can be organized using e-beam lithography, while for covering large areas, it can be crystallized in solution and deposited or crystallized on the surface. DNA-assisted lithography (DALI) method combines the use of DNA origami with the robustness of conventional lithography to create discrete, well-defined, and entirely metallic nanostructures with designed plasmonic properties. The method uses a transparent sapphire/silicon nitride chip where an amorphous silicon layer is grown on top, then a treatment with oxygen plasma is applied followed by the drop-casted of DNA origami nanostructures on the chip. Next, a silicon dioxide (SiO_2_) layer mask is selectively grown and DNA origami-shaped silhouettes are left on the layer, then the silicon is etched away by reactive ion etching (RIE). This step is followed by a physical vapor deposition (PVD) of a gold film, and the elimination of silicon dioxide and the remaining silicon, results are discrete origami-shaped metal nanostructures [251].

#### 4.1.7. 3D DNA Nanostructures

DNA is a precisely programmable material appropriate for directing well-defined three-dimensional (3D) arrays. This kind of array is possible by the variation of building blocks (inorganic and bio-organic composition, size and shape), assembly methods (ssDNA, DNA tile, DNA frame origami nano-structure), and dynamic manipulability (design, chemical stimulus, physical stimulus). Several methods have been reported for the creation of distinct 3D ordered arrays (diamond type, simple cubic, and body-centered-cubic) with the combination of 3D DNA frames (with shapes such as tetrahedron, octahedron and cube), metallic and or semiconductor nanoparticles and proteins [252]. Additionally, the successful connection of DNA frame origami structure for 1D and 2D arrays is possible to extend low-dimensional arrays to 3D lattice through the knowledge of driving factors of dynamic assemblies of 3D lattice [253].

The polyhedral DNA frames with encapsulated nano-objects are called DNA material voxels and they can be coupled with proteins, NPs and QDs. 3D DNA lattices made with voxels with different symmetries show desirable characteristics such as diffraction-limited spectral purity, light-emitting, 3D packaging of desired nano-objects, and the possibility of manipulating and enhancing enzymatic cascade reactions by the lattice architecture [252,253]. Additionally, polyhedral DNA origami can be used as a frame for fabricating gold nanoparticle clusters with high spatial complexity arrangements that can mimic the position of atoms in a crystal lattice unit cell [254].

There are some examples of 3D lattices using designable DNA origami building blocks. DNA Origami building blocks arranged in a toroidal geometry assembled with AuNPs shows stronger chiroptical response with a specific circular dichroism spectrum. While opposite response was observed with the enantiomers of the plasmonic structures. This characteristic allows the use of this materials to create plasmonic platforms chiroptical response along their axial orientation for enantiomer sensing [13]. Therefore, those type of tailored nanostructured molecules exhibit potential applications of sensor area utilizing chemically-based assembly systems. 3D rhombohedral crystalline lattice nanostructures assembled with DNA origami based tensegrity triangles had the characteristic of being an open structure. These crystals are spacious enough to efficiently host 20 nm gold nanoparticles and they would also suffice to accommodate other macromolecules [80]. Fabrication of DNA origami lattices, using polyhedral or triangular DNA origami single monomers as the building blocks, as well as by co-assembling two different shapes of DNA polyhedra have less difficulties to construct compared with Wulff single crystals. DNA origami frames (DOFs) with programmable geometries and binding behaviors can be crystallized into well-defined Wulff single crystals (the shape with minimal surface energy for solids of arbitrary crystallographic symmetry) [253]. Some techniques have been reported to construct this kind of crystals, for example the silica-encapsulation strategy for DNA microcrystals and the crystallization of cubic microcrystals from R-octa or E-octa DOFs through a slow annealing process is achieved by the use of DNA sequences of sticky ends in the R-octa system [255]. The wide variety of 3D geometries as well as their composition have been used to create molds for the nanocasting process of nanoparticles. This method utilizes seeded-growth nanoparticle synthesis with the seed beginning within a 3D shaped cavity made with DNA that allows the growing nanoparticle will replicate the cavity. Designed structures exhibited plasmonic properties with diverse inorganic materials, offering a range of applications in biosensing, photonics, and nanoelectronics [256]. The dimensions of structures that have been achieved with DNA have reached the micrometer-scale. Through hierarchical assembly of DNA origami blocks via shape complementarity, sticky-end association and multi-stage assembly have made possible enormous gigadalton-scale ordered structures, and thus brings possibilities to bridge molecular and macroscopic scales [257]. Three-dimensional plasmonic nanoarchitectures construction represents a challenging task in nanotechnology that DNA origami techniques had somehow overcome, which allows us to engineer complex plasmonic shapes with tailored optical response.

**Table 3 biosensors-12-00962-t003:** Examples of applications of nucleic acid based metamaterials.

Type	Configuration	Method	Optical Properties	References
1D	QDs attached to functionalized DNA origami nanotubes	**(a)** Self assembly of commercial streptavidin coated QDs on pre-engineered DNA nanostructures that display biotin molecules at selected locations. **(b)** Hybridization of DNA functionalized QDs to DNA structures carrying capture strands of complementary sequences.	Broad absorption but narrow and symmetric photoluminescence emission spectra, high quantum yield, excellent photostability, and resistance towards chemical degradation. Excellent properties for bio imaging.	[258]
	DNA-Gold Nanorods (GRN) films	Covalent conjugation of the thiolated gene of enhanced GFP to gold nanorods for the remote control of gene expression in living cells.	Optical switch that allowed induced enhanced green fluorescent receptor expression in HeLa cells after laser exposure.	[259]
	DNA Nanowires	Electrochemical synthesis of CdSe NCs with two different ssDNA molecules of 30 base guanine (poly G (30)) and 30 base cytosine (poly C (30)) as templates.	High GXRD peak intensities, excellent optical absorption and control of optical activities, as well as strong phonon confinement.	[260]
	DNA-CTMA fibers	Doping of lanthanides chelates into the DNA-CTMA matrix.	Fluorescence and optical amplification properties at suitable pumping wavelengths (612 nm) and low power.	[222]
	DNA-NP chains	Adjustment of the number of DNA modified on NPs and control of the assemblies through the design of the hybridized configuration to form dimers, trimers, pyramids, core-satellite, and chains.	Enhancement of EM fields, including SERS and chirality, for the amplification of optical properties.	[261]
2D	DNA nanoantennas	Consists of the attaching one or two nanoparticles (Au or Ag) to DNA origami structures with available docking sites for a single fluorescent dye next to one NP or in the gap between two NPs.	Provide the possibility of increasing fluorescence by plasmonic effects in the near-field of metal nanostructures more than 100 times the signal of the fluorophore.	[245]
	DNA nanorulers	DNA origami functionalized by qPAINT method to have a precise spatial control over number, spacing, and arrangement of fluorophores.	Have been used to evaluate the performance of TIRF microscopes and have the potential to be traceable ubiquituous standards for length measurements.	[242]
3D	Voxels	DNA frames with encapsulated nano-objects were called DNA material that can be coupled with proteins, NPs and QDs.	3D DNA lattices that show diffraction-limited spectral purity, light-emitting, 3D packaging of desired nano-objects, and the possibility of manipulating and enhancing enzymatic cascade reactions by the lattice architecture.	[252,255]
Others	Biodots	Fluorescent nanoparticles (biodots) of several nm in size synthesized from polymeric and monomeric nucleic acids: DNA, RNA, nucleotides, and nucleosides. Individual nucleotides are more suitable for biodot synthesis compared to DNA and RNA.	The fluorescence of nucleic acid biodots is stable in a broad range of pHs and in the presence of physiologically relevant cations. Applications include bioimaging and sensing platforms.	[198]
	Photoresponsive DNA-cross-linked hydrogels	Incorporation of Azo into the backbone of crosslinker DNA sequences give a Hydrogel phase transition regulated by UV/Vis irradiation.	Under visible light, the Azo molecule was in the trans form and allowed crosslinker DNA to hybridize with DNAs on the polymer side chains forming a 3D hydrogel network. When the gel was irradiated with UV light, the Azo was photoisomerized to the cis form that prevented hybridization and caused the hydrogel to revert to the sol state.	[262]

### 4.2. RNA Based Metamaterials

Ribonucleic acid (RNA) molecules are present in living organisms and some viruses and can have a variety of functions, lengths, and structures. RNA has various roles, such as structural, in gene expression processes (transcription, splicing, translation regulation), catalytic activity (ribozyme), and specific binding (aptamer), among others. When the human genome was sequenced, it was discovered that only 1.5% of the DNA codes for proteins, while the remaining 98.5% (popularly believed to be “junk DNA”), codes for non-coding RNAs that play a critical role in the regulation of cellular functions [263]. RNA has several unique attributes that make it a powerful biomaterial compared to DNA. The anionic polymer RNA is more prone to forming intrastrand double helixes and diverse tertiary and quaternary structures than DNA. RNA itself tends to have strong and complex secondary and tertiary structures by canonical and non-canonical base pairings, which enables the assembly of unique RNA nanostructures. RNA molecules can fold into a great diversity of structural motifs and interactions (Figure 6a,b), stabilized by tertiary interactions and complex 3-dimensional architectures exhibiting pseudoknots, single-stranded loops, paranemic motifs, kink-turns, bulges, three-way and multi-helix junctions, hairpins, kissing loops, pRNA multimers, base stacking and formation of triplexes and quadruplexes [264,265]. For the construction of RNA nanostructures (Figure 6), small structural motifs have been used to build a variety of 2D and 3D nanostructures with functionalities, such as nanoparticles, bundles, membranes, polygons, arrays, and microsponges that have potential applications in biomedical and material sciences [266,267,268].

The first demonstration of the feasibility of RNA nanotechnology was published in 1998 by Guo with RNA dimer, trimer, and hexamer nanoparticles constructed through self-assembly of packaging RNAs (pRNA) from the bacteriophage phi29 DNA packaging motor molecules [269], as it can be see at Figure 6d. Then the discovery of the ultra-thermostable RNA motif 3WJ (three-way junction) raised the number of publications on RNA applied to nanotechnology. 3WJ motifs are distributed in ribosomes, ribozymes, and other RNAs. Bacteriophage phi29 pRNA 3WJ has been used as a block to construct multi-functional nanoparticles with therapeutic effects due to their thermodynamic stability [270,271]. The 3WJ scaffold can be modified through its branch extension to form architectures such as X-motif or dendrimer-like 3D globular structures. The 3WJ can also be a building block to form 2D planar polygonal structures by stretching the intrahelical angle to form triangle, square, and pentagon pRNA nanoparticles. Additionally, 3D RNA nanoparticles can be assembled, such as tetrahedron and prism architectures. pRNA allows the incorporation of functional modules such as aptamers, siRNAs, miRNAs, DNAzymes, fluorogens, RNA-based fluorogenic modules, and many molecules without affecting the folding of the central pRNA core (Figure 6c). Although RNA origami techniques are underdeveloped because of their high cost, several RNA origami structures by assembly of bricks, scaffolds, or single-stranded are being developed continually by synthesizing long RNA strands with enzymatic transcription [272]. RNA origami nanostructures can be functionalized to promote the binding of nanostructures by aptamer-based ligand [273], for cargo encapsulation by RNA-protein interaction motifs or RNP motifs [274], for assembly and disassembly of nanostructures by enzymes that produce or degrade RNA [275], and RNA-RNA kissing interactions for higher order assembly [276].

On the other hand, the methodology for designing and constructing RNA nanostructures/nano objects through controlled self-assembly of modular RNA units is called tectoRNAs [277]. RNA tectonics has enabled the cotranscriptional production of sophisticated shapes, including a heart [278]. The most used motif for nanoarchitecture construction is the pRNA-3WJ; however, other RNA tertiary motifs have been used for various structures and applications [279]. For example, the HIV kissing loop structural motif has been used to construct tectoRNA nanoparticles in the shapes of squares [280]. Those squares can be assembled in two-dimensional and three-dimensional RNA square array structures (Figure 6). Programmable RNA nanocubes work as scaffolds for multiple functions; their configuration can be RNA-RNA, RNA-DNA, or dsRNA. Moreover, the conjugate of gold nanoparticles and structural RNA (tectoRNA) have the potential to become a new generation of therapeutics that have useful optical properties and biocompatibility advantages [281].

Advances in the folding and structure of RNA motifs and junctions have laid a foundation for further developing diverse RNA nanoparticles, as it can be observed with some examples in Figure 6. Some structures have been assembled, such as squares [280], jigsaw puzzles [282], filaments [283], cubic scaffolds [284], and polyhedrons [285]. 3D RNA nanostructures such as pRNA nanocages, tRNA polyhedrons, RNA cubes, nanoprisms, and RNA dendrimers are particularly useful for encapsulating drugs and imaging modules, and for the controlled release of the modules at specific sites [286]. Programmable RNA nanorings can regulate the formation and properties of silver nanoclusters (AgNCs) [287]. RNA also can form a membrane, a macroscopic structure without any polymer support or complexation with the potential to control drug-release [288].

Other interesting materials are the RNA hydrogels that can be made of RNA-triplehelix coupled with therapeutic molecules [289]. Hydrogels with functional RNAs may also be constructed by combining traditional polymer hydrogel materials and RNA components, including RNA nanoparticles [290]. RNA molecules with proper sequence motifs can self-assemble into a 3D hydrogel polymer network without the addition of any cross-linkers or any external or artificial support [291]. Films made with tRNA showed linear and nonlinear optical properties, representing a high potential in all-RNA photonic device applications and further applications to temperature sensing and thermo-optic devices [292].

### 4.3. DNA Production

DNA and other nucleic acids are increasingly considered promising biomass resources. DNA is a natural product that is safe for humans and the environment. DNA functionalization is possible due to its ionic character, and it can be performed by electrostatic interaction, intercalation, and statistical doping [223]. Next, a description of the most important methods for obtaining nucleic acids is made, either by in vitro or in vivo synthesis, as well as through their isolation from natural sources. Figure 7 shows a summary of the methodologies analyzed in this paper.

### 4.4. DNA Synthesis

DNA synthesis has become an enabling technology for modern DNA-based biomaterials, a broad applicability in nanotechnology. There are methods in vitro and in vivo for DNA synthesis, and the available approaches can be divided into chemical synthesis, enzyme synthesis, and bacteria-based synthesis. Chemical synthesis is a method that can generate ssDNA without templates [293]. Enzymatic synthesis includes the production of DNA by ligation or polymerization reactions [10,294]. A bacteria-based synthesis is a scalable approach based on fast-growing *Escherichia coli* (*E. coli*) cells and bacteriophages, and it offers mg-scale yields of ssDNA [295]. Other in vivo methodologies are also emerging (Retrons and RC-replicating plasmids).

#### 4.4.1. In Vitro Methods

##### **(A) Chemical Synthesis** 

Single-stranded DNA fragments of less than 200 nt are mainly produced by direct chemical synthesis. The synthesis of short oligonucleotides is usually performed by different kinds of phosphoramidite chemistry methods. Chemical synthesis is a cyclical process that elongates a chain of nucleotides from the 3′-end to the 5′-end based on standard phosphoramidite chemistry. It consists of a four-step chain elongation cycle: deprotection, coupling, capping (optional), and oxidation [296,297]. The fully synthesized sequence is chemically cleaved from the solid support, and the protecting groups are removed. The length of oligos synthesized by the phosphoramidite chemistry-based methods is limited to 200 nucleotides in general [298]. Chemical synthesis can be carried out using either column-based or microarray-based synthesizers. Additionally, short oligos can be assembled into long DNAs [299].

**Column-Based Oligo Synthesis.** The synthesis is carried out separately in columns where reagents are pumped, enabling the iterative addition of nucleotides in a programmable way [297]. Advances in materials, automation, procedure, and purification have led to an increase in production capacity and purity. The commercially available column-based oligo synthesizers usually synthesize 96–768 oligos simultaneously at scales from 10 to 2 μmol [300]. However, this method cannot satisfy the requirements of large-scale DNA synthesis due to the limitations of low throughput and high cost [296,297].

**Microchip-Based Oligo Synthesis.** The synthesis is carried out separately in columns where reagents are pumped, enabling the iterative addition of nucleotides in a programmable way (Kosuri and Church, 2014). Advances in materials, automation, procedure, and purification have led to the synthesis and increased production capacity and purity. The commercially available column-based oligo synthesizers usually synthesize 96–768 oligos simultaneously at scales from 10 to 2 μmol [300]. However, this method cannot satisfy the requirements of large-scale DNA synthesis due to the limitations of low throughput and high cost [296,297].

**Microchip-Based Oligo Synthesis.** It consists of a large-scale oligo parallel synthesis on a silica surface and provides an inexpensive source of oligo building blocks for various applications. The synthesizers are based on the principle of phosphoramidite chemistry with slight modifications in the steps of deprotection and base coupling [296]. Synthesizers are evolving and adopting new technologies such as light control [293,301,302,303], electrochemical [304], and inkjet printing methods [193]. These methods have enabled the high-fidelity synthesis of oligo of around 300 nt with a cost 2–4 orders of magnitude cheaper than the column-based oligo synthesis [297].

##### **(B) Enzymatic Synthesis** 

In contrast to chemical synthesis, enzymatic methods can reduce the formation of by-products and the depurination of DNA and other damages, and longer oligonucleotides can be synthesized directly. The length of ssDNA fragments synthesized ranges in size from several hundred base pairs to several thousand bases. Several methods based on enzymatic synthesis have been reported. The template-independent method involves the mechanism of terminal deoxynucleotide transferase (TdT)-based ssDNA synthesis, while the methods with polymerase-dependent are in vitro transcription and reverse transcription (ivTRT), the mechanism of asymmetric polymerase chain reaction (aPCR), the primer exchange reaction (PER) cycle and the rolling circle amplification (RCA).

**TdT-based ssDNA synthesis.** The main activity of TdT is the addition of multiple random deoxynucleotide triphosphates (dNTPs) to the 3′ end of an ssDNA [305]. To overcome problems with the control of adding single bases, a method of light-mediated deprotection synthesis using dNTPs with 3′-blocked groups has been created [306]. Because TdT has many limitations with 3′-blocked dNTPs and results in low extension yield, another method was developed, and the TdT was conjugated to a dNTP molecule [298].

**ivTRT.** This method uses a dsDNA template (PCR product or plasmid) to synthesize RNA via transcription, then the RNA is used as a template to synthesize an ssDNA using reverse transcriptase, and the RNA template is finally cleaved using RNase H [307,308]. It can be used to obtain ssDNAs of about 0.5∼2 kb in length. However, ivTRT is labor intensive and expensive, and using nucleases can limit the product yield and requires DNA of high quality [309].

**aPCR.** Synthesis of ssDNA from a dsDNA template to generate ssDNAs ranging from hundreds to thousands of nucleotides. It consists of using two amplification primers in unequal concentrations and two phases of amplification for dsDNA templates exponential amplification and linear amplification of ssDNA [310]. Once the limiting primer is fully used, the excess primer extends to form ssDNA during the rest of the cycles. DNA for origami scaffolds of custom length and sequence of up to kb scale has been obtained [311,312]. Additionally, ssDNA synthesized by aPCR was used to fold DNA nanoparticles of diverse shapes and sizes, whose sequences were designed by software that converts any 3D solid object into the synthetic DNA sequences needed to synthesize the target object [311].

**RCA.** It consists of the isothermal amplification of a circularized ssDNA template mediated by a polymerase with strand displacement capabilities (Phi29 (*ϕ*29) polymerase purified from the *Bacillus subtilis* bacteriophage *Φ*29). The polymerase begins extending the primer around the template to form a complementary strand until the starting point is reached; then, the polymerase displaces the newly synthesized strand and continues the synthesis. It generates a concatemeric ssDNA, and the process can be maintained for up to 8 h with a yield of up to milligram quantities of pure ssDNA [313]. To generate smaller fragments, some strategies have been implemented to cut the ssDNA. It has been achieved by the coupling of complementary oligos to ensure that restriction enzymes can cut the sequence [314]. Other strategies are based on designing the initial sequence so that once ssDNA has been synthesized, it folds into hairpins where enzymes can cut [315]. DNAzymes have also been incorporated [316]. Despite the high simplicity and the large amount of ssDNA produced by RCA, the use of concatemeric ssDNA scaffolds is limited to nanostructures due to the repeated motifs. This sequence limits the monodispersity of the assembled architectures and does not allow discrete nanoparticle assembly [317].

**PER.** It consists of the programmed sequential extension of seed ssDNA through multiple DNA hairpin primers/templates with the Bst DNA polymerase from *Bacillus stearothermophilus*. Each newly extended product released from the hairpin can serve as a primer in another hairpin in the next PER cycle [318]. This method allows arbitrary sequences to be concatenated into a desired transcript, and reactions can be cascaded together because they follow a designed growth pathway. DNA origami has been constructed by the generation of 40 staple oligonucleotides with the reaction of 40 primers with 80 different catalytic hairpins [319].

#### 4.4.2. In Vivo Methods

**Bacteriophage-Based ssDNA Production.** A common, cost-effective source of ssDNA is the circular genome of the filamentous bacteriophage M13 that can be deftly and easily engineered. M13 infects *Escherichia coli* and then replicates to produce more phages, which are released directly into the culture medium without bacterial lysis. The genome from progeny phages is then purified and used as a source of ssDNA. The DNA genome from bacteriophage lambda of approximately 48.5 kb has been used to fabricate nanowires with gold nanoparticles [320,321]. Additionally, the M13 bacteriophage genome is utilized in ’DNA origami’ for assembling nanostructures of complex geometries [322]. An engineered version of the bacteriophage M13, called M13mp18 [323], offers a higher replication rate, so this sequence or variants of this have been used as a scaffold for DNA origami [324,325,326]. A high yield of ssDNA has been obtained with the control of culture conditions to reach high-density bacterial culture and the control of other variables, such as the ratio of phage used to infect a culture and the time of infection [327]. With M13mp18, the size of the ssDNA scaffolds produced has many limitations. Phagemids are a good alternative to bacteriophage infection. Phagemids can be replicated as plasmids (dsDNA) or as circular ssDNA due to the presence of a second origin of replication that comes from f1 or M13 bacteriophage. In contrast to M13mp18 strategy, phagemids do not encode for any viral protein, but another plasmid called “helper” has the viral information to package the ssDNA encoded by the phagemid [328]. This method with variations of phagemids and helpers has been applied to create ssDNA scaffolds from different sizes and folded into DNA origami structures [329,330,331,332,333]. Later, hybrids with bacteriophage λ generated an ssDNA scaffold of 51,466 nucleotides [334]. Additionally, phagemids that simultaneously encode the scaffold and staple ssDNA have been useful for assembling DNA nanorods [335]. Despite the high yield of produced ssDNA by the phagemid method, a common problem is the presence of plasmid dsDNA contaminants [336]. Through bacteriophage-based ssDNA production, DNA scaffolds from 1317 to 51,466 nucleotides at a scale of Shaker flask and bioreactor have been reported [317].

On the other hand, the need for scalable methods to produce pure ssDNA scaffolds with custom lengths and sequences is becoming crucial as the DNA origami field continues to grow rapidly. DNA origami requires single strands of DNA of virtually arbitrary length and with virtually arbitrary sequences, and they can be produced in a scalable and cost-efficient manner by using bacteriophages through the generation of single-stranded precursors of DNA. First, the excision of individual oligonucleotides was reported from a circular single-stranded DNA precursor with type IIS restriction endonucleases [315]. Then, the target strand sequences were interleaved with self-cleaving catalytic activity DNAzymes. Since DNAzymes are part of the same DNA strand as the target oligonucleotides, they are mass-produced together with the target DNA, and it enables efficient production without the need for additional costly components. In this last method, *Escherichia coli* cultures along with a bacteriophage system were used to obtain the phagemid DNA in a single-stranded form through rolling circle amplification, followed by packaging into extracellular phagemid particles. Phagemid ssDNA is then isolated from phagemid particles and cut by the activation of the DNAzyme with ZnCl_2_. Finally, the staple molecules are recovered and can be used for self-assembly reactions to mass-produce the designed target shape. With this strategy, by genetically encoding both scaffold and staple strands, it is enough to have seven liters of culture to yield a gram of DNA origami covering a surface of 1000 m^2^. With an 800-liter scale, total costs could be approximately USD $200 per gram [335].

**Retrons.** Retrons, an anti-phage defense in bacteria, have also been used for ssDNA production in vivo. Retrons are genomic DNA, a retroelement that produces multicopy single-stranded DNA (msDNA) that codes for reverse transcriptase and a hybrid of ssDNA/RNA [337]. Interestingly, with the co-expression of a recombinase, the intracellularly expressed ssDNAs can introduce precise mutations into genomic DNA, thus transforming transient cellular signals into genome-encoded memories [338,339]. Variations in the retron architecture have a direct relationship with ssDNA production [337]. ssDNAs from 32 to 205 nt were synthesized by retrons and either assembled into DNA nanostructures in vivo or purified for in vitro assembly [340].

**RC-replicating plasmids.** Plasmids are self-replicative DNA elements that are transferred between bacteria. Apart from encoding antibiotic resistance genes, they could also encode adaptive genes. Thousands of plasmids use the rolling-circle mechanism to propagate (RC plasmids) [341]. The RC-replicating plasmid pC194 from Gram-positive bacteria can replicate and produce circular ssDNA in *E. coli* [295,342].

#### 4.4.3. DNA from Natural Sources

DNA synthesized by in vitro or in vivo systems comprises custom DNA sequences, oligonucleotides, and longer constructs, such as synthetic genes and even entire chromosomes [293]. Synthetic DNA with a specific design is still too expensive and not easy to produce in large quantities to be feasibly translated to an industrial scale. This makes it difficult to apply the synthetic DNA to materials requiring much larger quantities. Therefore, DNA from natural resources has been explored to construct nanostructures.

DNA available from natural sources has substantial potential to be mass-produced for its use in production of materials [343]. Among available biomass feedstocks, nucleic acids (e.g., DNA, RNA) represent ubiquitous biomacromolecules, readily available at a reasonable cost from fish milt and gonads, plants, yeasts, and other renewable sources. Biomass DNA comprises only about 3.1% of dry weight of bacteria, 0.1–0.6% of yeast and 1% in mammalian cells [344]. Salmon milt in fisheries, which is often used for livestock feedstuff, contains over 10% dry weight DNA, and it is very easy to manufacture [345,346]. Salmon fishing industry has a worldwide supply exceeding 2.4 million tons per year, with an estimate of about 3000 tons of salmon DNA available per year [347,348,349]. DNA is processed from the salmon roe and milt sacs to obtain highly purified DNA, suitable for optical device fabrication [350]. Other sources of cellular biomass already exist in large scale from industry settings such as fermentation wastes (dregs and residues), food process waste (thymus, spleen, pomace, etc.), or from the environment such as bloomed algae [351]. DNA that comes from renewable resources is inherently cost effective and it will effectively promote further DNA research in a variety of scientific fields.

There are several reports about the utilization of biomass DNA as a functional material. Despite the variability in length of DNA chains from crude DNA material, the features of this kind of DNA, such as chirality, negative charge, periodicity and diameter, enable it to be used in the fabrication of complex structures when combined with versatile functional materials [352]. Salmon sperm DNA duplex has an average length of 2000 bp with a corresponding contour length of around 680 nm and is regarded to be a semi-flexible polymer with a persistence length of ∼50 nm. This kind of DNA has been used to form liquid crystal (LC) phases amongst other applications [352]. On the other hand, DNA from different species has been converted into diverse and useful materials including hydrogels, organogels, composite membranes, and plastics [343]. DNA attached to cellulose (DNA-filter hybrid) has been successfully employed for the recovery and separation of REE (rare earth elements) [353,354]. DNA-films were found to selectively remove harmful chemical compounds with planar structure, such as DNA intercalating compounds and endocrine disruptors [355]. Thanks to DNA versatility and thin film processability, it has the possibility of tailoring optical and electrical properties [223]. DNA functionalized with the hexadecyltrimethylammonium chloride (CTMA) surfactant shows excellent film formation and light propagation properties, and is thermally stable up to around 230 °C [356]. DNA-CTMA films have been incorporated as intermediate layers in many optoelectronic devices [357]. DNA-CTMA films are particularly efficient at intercalation with various dyes, which makes it useful for nonlinear optics [223], laser technology [228], and electrochromic devices [358]. Furthermore, differences between chromosomal and plasmid DNA have been explored in terms of their stability with the CTMA complex when a monolayer is transferred onto a solid substrate. Plasmid showed better stability, suitable for applications in organic electronics [229]. Additionally, DNA extracted from salmon milt is a viable bio-template to prepare stable nanoparticles [359,360].

Commercially available genomic DNA has been commonly used as a template to assemble nanowires without further modification as it is inexpensive and widely available. The polyanionic properties of DNA allow the adsorption of cationic species, while the template nucleotide sequence itself is largely irrelevant [224]. Other sources of genomic DNA, such as calf-thymus and salmon (with an average size ∼50 Kbp), have also been used for 1D DNA-templated structures [361,362]. Materials such as nanorod-DNA thin films and DNA-CTMA fibers were fabricated with salmon DNA, providing advantages in the selectivity of dimensionality, solubility in organic and inorganic solvents, and functionality enhancement [217]. Furthermore, salmon DNA was used in the assembly of films with optoelectronic characteristics and made of synthetic double-crossover (DX) DNA lattices and the doping of double divalent metal ions [363].

### 4.5. RNA Production

For the use of RNA in the production of nanomaterials, large amounts of RNA are required. RNAs are generally produced by labor- and cost-intensive in vitro transcription (enzymatic) or via chemical synthesis (automated solid-phase synthesis), which are costly and difficult to operate on a large scale or limit the length of the product (Figure 7). Compared to in vitro approaches, in vivo expression is cheaper and much more practical on a large scale. Additionally, this type of genetic material can also be obtained from natural sources as we will review later.

#### 4.5.1. In Vitro and In Vivo Synthesis Methods for RNA

Solid-phase chemical synthesis allows the addition of modifications, the purification is simple, and is a fast procedure. However, the equipment required is expensive, the length of the product is limited to ∼100 nt, and the number labels or modifications available is highly limited and carries a high cost [364,365].

The mechanism of in vitro transcription allows the template-directed synthesis of RNA molecules of any sequence from short oligonucleotides to several kilobases in μg to mg quantities [366] (Beckert and Masquida, 2011). Standard protocols use a template that includes a strong bacteriophage promoter upstream of the sequence of interest and the product is purified by preparative, denaturing polyacrylamide gel electrophoresis (PAGE). The most commonly used promoter is T7 and transcription is mediated with the corresponding RNA polymerase. Enhancement of RNA yield was reported in a modified in vitro transcription protocol, by adjusting incubation temperature and limiting GTPs [367].

Alternatively, large RNA scale production could be possible by fermentation in bacteria, because It offers a cheap and rather simple alternative to in vitro synthesis. DNA templates can be incorporated into a plasmid and transformed into *E. coli* to take advantage of the cellular transcription machinery. The genetic information inside the plasmid will be transcribed into the corresponding ssRNA, which spontaneously folds into the designed nanostructures inside the cells. Specific tags that usually encode an aptamer that recognizes a specific ligand are usually used for affinity purification [368].

For the tRNA-scaffold approach, the RNA sequence of interest was hidden within a standard tRNA structure (tRNA-scaffold), this arrangement allows for the recombinant RNA to escape ribonucleases. It is later processed by the standard *E. coli* tRNA processing enzymes and accumulates in copious amounts inside the bacterial cells [369]. Although this approach yields large amounts of pure and homogeneous recombinant RNA (around milligrams per liter), the RNA fragments need extra purification steps [368]. Other authors have used another kind of scaffold such as the 5S ribosomal RNA with excision of the product RNA mediated by a biotinylated DNAzyme cleavage [370,371]. As ribosomes are metabolically expensive to produce in the cell, and ribosomal RNAs, unlike other RNAs such as mRNAs, are resistant to degradation and have long half lives (days) in the cell [372].

New approaches for RNA synthesis try to take advantage of the characteristics of circular RNAs, a unique type of noncoding RNA molecule that has been found in a wide range of cells and is a covalently closed ring produced by a process called backsplicing [373]. The structural advantage of circRNAs is its higher stability against degradation by exonucleases [374]. In vitro methods for the circularization of artificially- produced RNA are being developed [375].

#### 4.5.2. Purification of RNA from Natural Resources

The extraction of RNA from natural samples requires a purification process to separate it from the rest of the cellular components. Selective isolation of RNA can be achieved by solvent extraction and precipitation methods that take advantage of differential solubility of biomacromolecules in different solvents and ionic conditions. Solvent extraction to eliminate proteins and DNA components followed by precipitation is the method of choice to isolate large amounts of total RNA from natural sources [376]. RNA content can vary widely between tissues, cell-types, physiological state, among others. Tissues where RNA is plentiful, such as liver, spleen and heart obtain a yield of 2–4 μg of total RNA per mg of tissue [377]. A typical mammalian cell contains 10–30 pg total RNA [378], where the majority of RNA molecules are rRNAs and tRNAs. While mRNA accounts for only 1–5% of the total cellular RNAs, it depends on the cell type, cell cycle state, cell size, aging, and environmental conditions [379]. RNA isolated from plant tissues gets from 0.031 μg to 1.8 μg of total RNA per mg of tissue [380,381,382]. The high content of starch, proteins and fiber in plants affect the purification [382]. To obtain higher purity, some techniques can be performed such as ultracentrifugation, PAGE and chromatography [365]. Ultracentrifugation with a gradient on top of a sucrose cushion is the standard way of purifying large RNA structures such as ribosomes, polysomes, or individual ribosomal subunits from lysed cells [383]. Another method to purify large amounts of RNA (microgram to milligram scale), which is easily applied to a wide range of sizes with single-nucleotide resolution is by polyacrylamide gel electrophoresis (PAGE). RNA of interest is obtained by excision of the band of interest from the gel, followed by electroelution or crush and soak extraction [384]. For liquid chromatography, the separation of solutes is achieved through the passage of the solute contained in a mobile phase by a column containing a solid material (stationary phase) that interacts with the solutes to a different extent [365]. By Reversed-phase ion-pairing chromatography, RNA is retained by the interaction with a lipophilic cation-pairing agent [385]. By Ion-exchange chromatography, the stationary phase is positively charged to allow the interaction with the negatively charged RNA strands [386]. While in the case of Affinity chromatography, the stationary phase is functionalized with poly uridine (polyU) to interact with polyA tails of messenger RNAs (mRNA) [387]. The Size-exclusion chromatography can separate large RNAs that elute first from the stationary phase whilst smaller RNAs are absorbed into the porous medium [388].

RNA is also produced industrially from living organisms, for example, to obtain RNA degradation products that are commonly used for condiments, such as guanosine monophosphate (GMP), an active flavor enhancer [389]. *Candida tropicalis* is widely used in industries to produce RNA for its higher RNA content (166 μg of total RNA per mg of dry weight), but there are many concerns about its safety [390]. Another source of RNA is *Saccharomyces cerevisiae*, a safer microorganism that is easy to cultivate with high yield of RNA (59 μg of total RNA per mg of dry weight) [391,392]. Very few advances have been made in the use of biomass RNA as a component of nanostructures. There is some commercially available biomass RNA, for example RNA from bacteriophage MS2, RNA from yeast, RNA from Torula yeast, tRNA from wheat germ, human 28 s ribosomal RNA among others. There are still very few applications towards the field of optics where RNA derived from biomass has been used. The tRNA powder processed from wheat germs has been used for the assembly of RNA membranes [292]. Nanoparticles were also obtained from *E. coli* tRNA with divalent metal cations, which made it possible to obtain particles ranging in size from several to hundreds nanometers [393]. Finally, RNA–CTMA complex film was prepared by spin-coating it onto an indium tin oxide (ITO), a promising candidate for electronic devices [4]. Surely with the increase in the applications of this type of genetic material, there will also be improvements in RNA production options.

## 5. Conclusions

The utilization of biomolecules to develop nanostructured metamaterials offers an improved design flexibility, and expands their physical, chemical, and biological properties, which are critically important for their application in the optical biosensor field. Combining chemical compounds with biomolecules has paved the way to the creation of nanoscale machines that are able to change their conformation, both tunably and reversibly, depending on the wavelength applied to them. Interest in the design and develop of carbohydrate, protein, DNA- or RNA-based nanostructures metamaterials has increased lately since they offer significant features such as chirality, biodegradability and biocompatibility, and are easily moldable and available, making them critically important for their application in biomedical, food, engineering, and biosensor fields.

In particular, carbohydrate-based metamaterials face limitations such as low strength parameters, limited wavelength work, and laborious manufacturing methods, making their production expensive and difficult for scaling-up. In the case of protein-based metamaterials, it is important to highlight that a significant advantage is the conjugation of the molecular and functional properties of the protein involved in the design of the metamaterial to be used. In other words, protein-based metamaterials can acquire various capabilities such as affinity, fluorescence, iridescence, biocapture, interaction specificity, or programmed structural conformational changes from the proteins used to manufacture them. On the other hand, being able to apply protein engineering in the optimization of the protein element of protein-based metamaterials opens up new design possibilities.

Apart from their biological functions, nucleic acids are arising as a novel tool for creating new functional materials. This is due their ability to bind inorganic materials, diversity in secondary structures, compatibility with nucleotide modifications, great variety of molecules for functionalization, high customizability of the sequence, and the great diversity of nanostructures that can be assembled (1D, 2D, 3D) through both bottom-up as well as top-down methods. Although production of these biomolecules could be a bottleneck for the widespread use of them in the engineering of metamaterials, a wealth of methods have been proposed, from extracting DNA and RNA from widely available biomass to highly customizable synthesized sequences through chemical and enzymatic methods.

Many challenges remain as biomolecular nanotechnology transitions into viable applications where matters of great scale production, cost, purity, stability, delivery, among other aspects, are paramount. Hence, research on bio-based materials design and characterization has expanded with the aim of developing new ones that can overcome those limitations.

## Figures and Tables

**Figure 1 biosensors-12-00962-f001:**
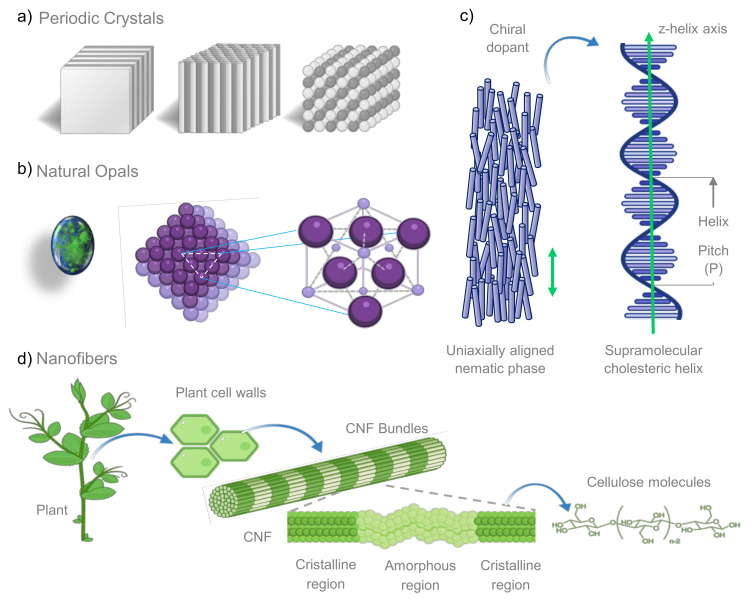
(**a**) One dimensional (1D), 2D, and 3D organization of photonic crystals, found in nature through the wings of several insects such as butterflies and cicadas. (**b**) Natural opals, an instance of periodic crystals, display a colorful aspect as a consequence of periodic systems with spheres substantially packed, which form a face-centered cubic lattice. (**c**) When nematic liquid crystals are doped with chiral molecules they can form supramolecular helix structures, conferring them remarkable features as diffraction grating, and selective light reflection. It is important to mention that the helix pitch is determined by the chemical structures of molecules forming the cholesteric phase, as well as the concentration of chiral moieties present in the dopant component. (**d**) Cellulose, the major component of cell wall plants, is organized in nanofibers (CNFs) possessing chiral nematic structures and 1D nanostructures with crystalline and amorphous regions.

**Figure 2 biosensors-12-00962-f002:**
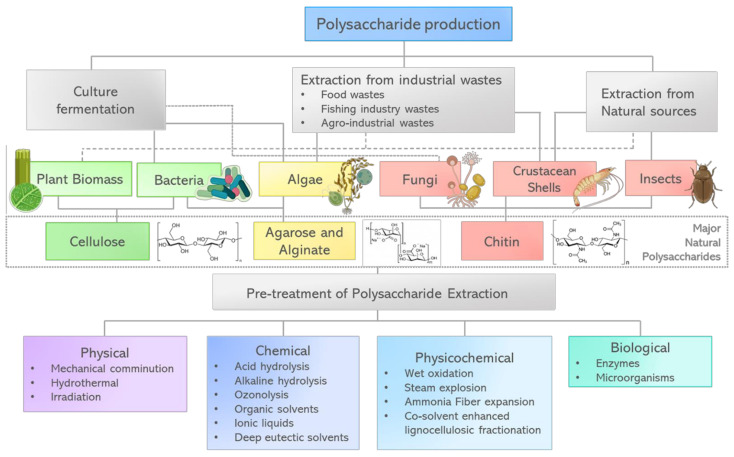
Major natural polysaccharides used in biomedical applications, highlighting their main sources, and the wide range of processes to isolate them.

**Figure 4 biosensors-12-00962-f004:**
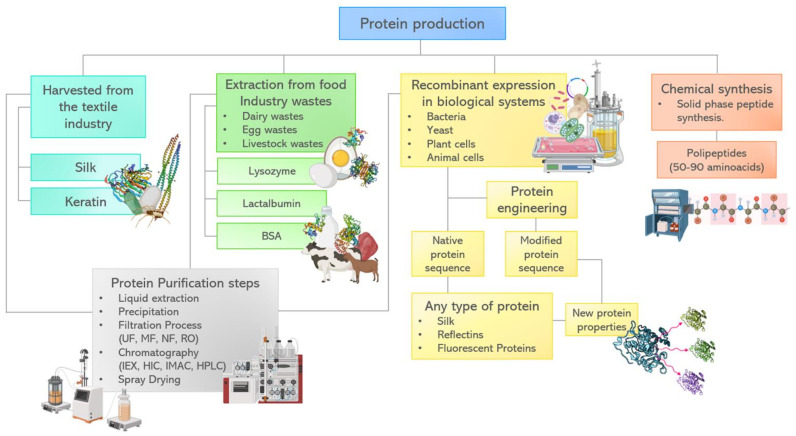
Protein extraction or production process alternatives and methods of synthesis.

**Figure 5 biosensors-12-00962-f005:**
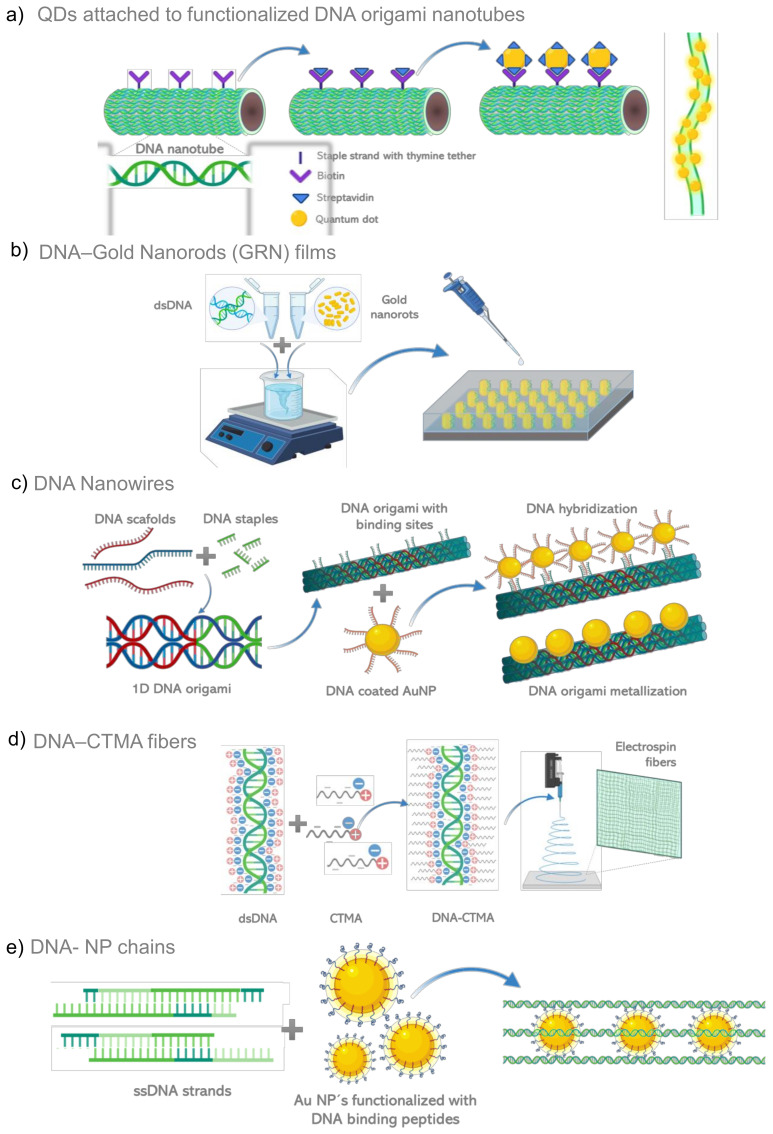
**One dimensional (1D) DNA nanostructures.**(**a**) Quantum dots attached to functionalized DNA origami nanotubes, (**b**) DNA gold nanorods films, (**c**) construction of DNA nanowires through the assembly of DNA origami and DNA coated AuNPs, (**d**) Electrospin fibers made of DNA-CTMA, (**e**) Formation of NP chains with the interaction between ssDNA strands and AuNPs functionalized with DNA binding peptides.

**Figure 6 biosensors-12-00962-f006:**
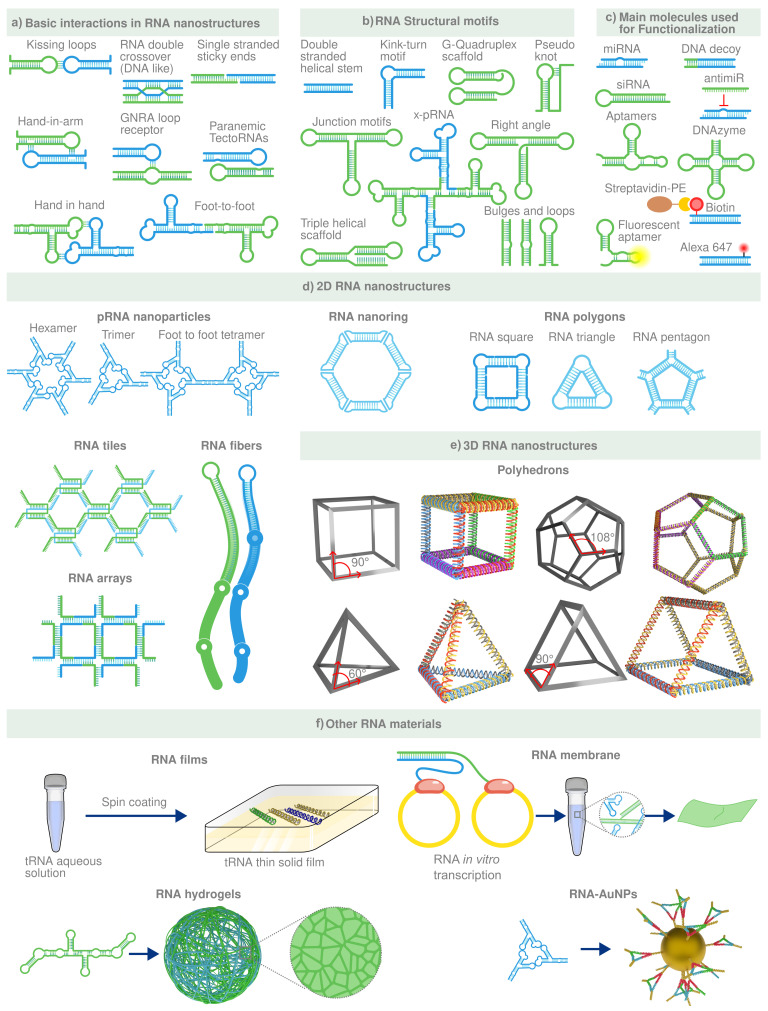
**Main aspect of RNA and the construction of nanostructures.** (**a**) Basic interactions between RNA motifs that are relevant for the assembly of RNA nanostructures. (**b**) Main RNA structural motifs. (**c**) Examples of molecules that are used for functionalization of RNA nanostructures. (**d**) Examples of 2D RNA nanostructures. (**e**) Main 3D RNA nanostructures. (**f**) Different kind of materials that have been created with RNA.

**Figure 7 biosensors-12-00962-f007:**
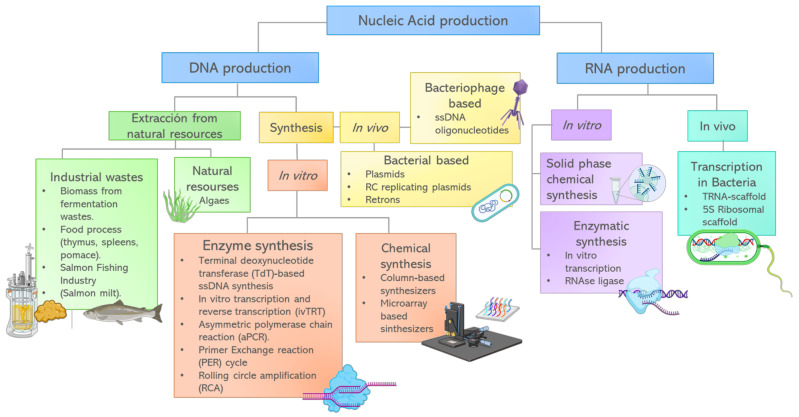
DNA and RNA sources from nature and methods of synthesis.

**Table 1 biosensors-12-00962-t001:** Carbohydrate based structured material and their optical properties.

Carbohydrate	Configuration	Method	Properties	References
Alginate	Optical films from Alginate/ POTE ^1^	Films were prepared directly from stock solutions of POTE and alginate through a solution-casting method. Mixed solutions were stirred for 12 h, placed into Petri dishes and vacuum-dried at 40 °C.	Tuning UV-visible absorbance and wettability behavior.	[32]
Agarose	Optical fibers	A melted agarose solution was poured into a glass mold with six internal rods. After cooling, the rods were removed to obtain air holes, and the solidified waveguide was released and cut.	Exhibit transmittance at 633 nm, which can be modulated for sensing purposes.	[33]
Carrageenan	Carrageenan-Ag Nanoparticles	A colloidal solution of carrageenan and AgNP was obtained by continuously stirring 1% *w*/*v* carrageenan in water and 50 mM AgNO_3_ for 24 h at 60 °C.	Dark brown to white shift color.	[34]
Cellulose	Biphasic nematic liquid crystal composite films	Obtained by combining cellulose nanocrystals with a low molecular weight nematic liquid crystal from HOB ^2^.	Iridescence under unpolarized room light.	[35]
Chitosan	MPA-Chitosan Quantum Dots (QD)	To prepare the QD, a solution of 1.25 mM CdCl_2_ and MPA ^3^ was poured into a flask under N_2_ atmosphere; in parallel a solution of NaHTe was obtained. Both CdCl_2_ and NaHTe solutions were mixed and subjected to a reflux at 100 °C under open-air conditions, thus obtaining water- compatible MPA-capped QDs. Finally, 3.56 nM QDs was added into 10 mL carboxymethyl chitosan solution, the mix was sonicated for 5 min, stirred, and vibrated overnight at room temperature in the dark.	Turn on-off fluorescence properties depending on the interaction with the analyte.	[36]
Chitosan	Planar optical waveguide from chitosan-Ag composite	Chitosan stock solutions were stirred with either a citric acid or an acetic acid solution, heated for 6 h, filtered and centrifuged. The chitosan solutions were spun onto the substrates in a spin coater. Later the films were air dried, and deprotonated by immersion into 3% ammonia solution.	Turning its refractive index through the silver ions reduction directly in controlled thickness chitosan films.	[37]
Starch	Starch-Ag nanocomposite films.	Fabricated via solution casting.	SPR at 438–449 nm.	[38]

^1^ POTE: poly (octanoic acid 2-thiophen-3-yl-ethyl ester). ^2^ HOBC: 40-(hexyloxy)-4-biphenylcarbonitrile. ^3^ MPA: 3-Mercaptopropyl acid.

## Data Availability

Not applicable.

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
