# Peer review of "Biomolecule-Based Optical Metamaterials: Design and Applications"

_biosensors, 2022, doi:10.3390/bios12110962_

Round 1
Reviewer 1 Report
Comments on the manuscript
In this manuscript entitled “Biomolecule-based optical metamaterials: design and applications” the authors have adequately reviewed the recent advancement in metamaterials constructed by biological components, such as carbohydrates, proteins, and DNA/RNA.
It is very impressive that the authors covered such a broad range of topics in a succinct manner. This review will be of interest to researchers in this field and also to future scientists and engineers who are preparing to get into this research field. Consequently, I support the publication of this review after a minor revision.
In the introduction part. “Of special interest are optical metamaterials that can alter electromagnetic waves at determined optical frequencies, conferring properties such as high transparency, high light absorbance, negative refractive index and hyperbolic dispersion [1,3].” Some related works on metamaterials absorbers are missing, for example, [Liang, Yao, et al. "Bound states in the continuum in anisotropic plasmonic metasurfaces." Nano Letters 20.9 (2020): 6351-6356].
Before my proposal of acceptance of this manuscript, I encourage the authors to expand the discussions to the direction of novel features of metamaterials/nanostructures enabled by biomolecules. I recommend some key references to help the authors.
1. Urban, Maximilian J., et al. "Plasmonic toroidal metamolecules assembled by DNA origami." Journal of the American Chemical Society 138.17 (2016): 5495-5498.
2. Kuzyk, Anton, et al. "A light-driven three-dimensional plasmonic nanosystem that translates molecular motion into reversible chiroptical function." Nature communications 7.1 (2016): 1-6.
3. Lin, Qing-Yuan, et al. "Building superlattices from individual nanoparticles via template-confined DNA-mediated assembly." Science 359.6376 (2018): 669-672.
4. Vázquez-Guardado, Abraham, et al. "DNA-Modified Plasmonic Sensor for the Direct Detection of Virus Biomarkers from the Blood." Nano Letters 21.18 (2021): 7505-7511.
Author Response
"Please see the attachment."

Reviewer 2 Report
The review draft by Torres-Huerta et al provides comprehensive and up-to-date analysis of recent recent applications of biomolecule-based metamaterials along with novel designs. The review can be considered for possible publication in Biosensors journal.
Author Response
"Please see the attachment."

Reviewer 3 Report
The authors have written a review article on optical metamaterials based on biomolecules, highlighting in particular their application in biosensing. For the most part, the review article is well structured and written in fluent English. I recommend that the submitted manuscript be accepted for publication after making minor changes, which are listed below:
· The introductory section is cpncise, but lacks important aspects of the review article. The authors are asked to provide a personal overview of why their review, along with many others on the topic of metamaterials, is important and how it differs from others
· What criteria guided the authors in selecting the scientific papers that included in this review? Did they use the specific keywords when searching the WoS database or it was some other criteria?
· It would be beneficial if the authors plotted a column bar or pie chart showing the number of papers published on the topic on metamaterials per year, or the distribution of papers in scientific journals by quartile.
· The authors are advised to review the entire manuscript in detail and to correct minor grammatical errors.
· It is recommended to put Chapter 2.2. before Chapter 2.1.
· Authors are requested use consistent citations throughout the entire manuscript, e.g., in lines 427, 434 and 448, cite immediately after the nameo f the research group leader. It is also not necessary to cite the same work in three consecutive sentences or in the same sentence, for example lines 254-256.
· It is recommended that long sentences that span multiple lines be split into two, e.g. lines 52-56 and others.
· Figures must be arranged to match the order in the text of the article, e.g., Fig. 3e (mentioned in line 431) is before Fig. 3c (line 510), before Fig. 3a (line 584) and before Fig. 3d (line 615). Therefore, Figure 3 needs to be reoredered to follow the numbering, i.e. to be consistent with the manuscript body.
· The tabular representations are somewhat dense, it would be nice to somehow reorder the method column, or at least add a separate column of references.
Author Response
"Please see the attachment."
